# Direct strain correlations at the single-atom level in three-dimensional core-shell interface structures

Hyesung Jo [1], Dae Han Wi[2], Taegu Lee[3], Yongmin Kwon[2], Chaehwa Jeong [1], Juhyeok Lee [1], Hionsuck Baik[4], Alexander J. Pattison[5,6], Wolfgang Theis [5], Colin Ophus [6], Peter Ercius [6], Yea-Lee Lee[7], Seunghwa Ryu [3], Sang Woo Han [2] ✉ & Yongsoo Yang [1] ✉

Nanomaterials with core-shell architectures are prominent examples of strain-engineered materials. The lattice mismatch between the core and shell materials can cause strong interface strain, which affects the surface structures. Therefore, surface functional properties such as catalytic activities can be designed by fine-tuning the misfit strain at the interface. To precisely control the core-shell effect, it is essential to understand how the surface and interface strains are related at the atomic scale. Here, we elucidate the surface-interface strain relations by determining the full 3D atomic structure of Pd@Pt core-shell nanoparticles at the single-atom level via atomic electron tomography. Full 3D displacement fields and strain profiles of core-shell nanoparticles were obtained, which revealed a direct correlation between the surface and interface strain. The strain distributions show a strong shape-dependent anisotropy, whose nature was further corroborated by molecular statics simulations. From the observed surface strains, the surface oxygen reduction reaction activities were predicted. These findings give a deep understanding of structure-property relationships in strain-engineerable core-shell systems, which can lead to direct control over the resulting catalytic properties.

By utilizing epitaxy between two different materials, the misfit strain at the interface can be finely controlled, and depending on the strain-induced structural change, the properties of materials can be dramatically altered[1–7]. Strain engineering techniques are now widely being studied and used in broad disciplines, allowing systematic optimization of optical, chemical, magnetic, and catalytic properties[2,5–10]. One of the prominent examples of strain engineering is the core-shell nanoarchitecture[4,5,7]. Notably, metallic core-shell nanoparticles can

show enhanced catalytic activities via interface strain effects and therefore have received great attention for their potential to be superior electronic catalysts for fuel cells[2,5–7,11,12]. The binding energy of adsorbates is especially influenced by strain[5,6,13,14], where it can be finely tuned by controlling the shell thickness or the core size[4,15–19].

To fully utilize the interfacial strain effect, it is essential to have an atomic-scale understanding of the 3D interface structure. However, studies to date have been mostly limited to ensemble-averaged

[1]Department of Physics, Korea Advanced Institute of Science and Technology (KAIST), Daejeon 34141, South Korea. [2]Department of Chemistry, Korea Advanced Institute of Science and Technology (KAIST), Daejeon 34141, South Korea. [3]Department of Mechanical Engineering, Korea Advanced Institute of Science and Technology (KAIST), Daejeon 34141, South Korea. [4]Korea Basic Science Institute (KBSI), Seoul 02841, South Korea. [5]Nanoscale Physics Research Laboratory, School of Physics and Astronomy, University of Birmingham, Edgbaston, Birmingham B15 2TT, UK. [6]National Center for Electron Microscopy, Molecular Foundry, Lawrence Berkeley National Laboratory, Berkeley, CA 94720, USA. [7]Chemical Data-Driven Research Center, Korea Research Institute of Chemical Technology (KRICT), Daejeon 34114, South Korea. ✉e-mail: sangwoohan@kaist.ac.kr; yongsoo.yang@kaist.ac.kr

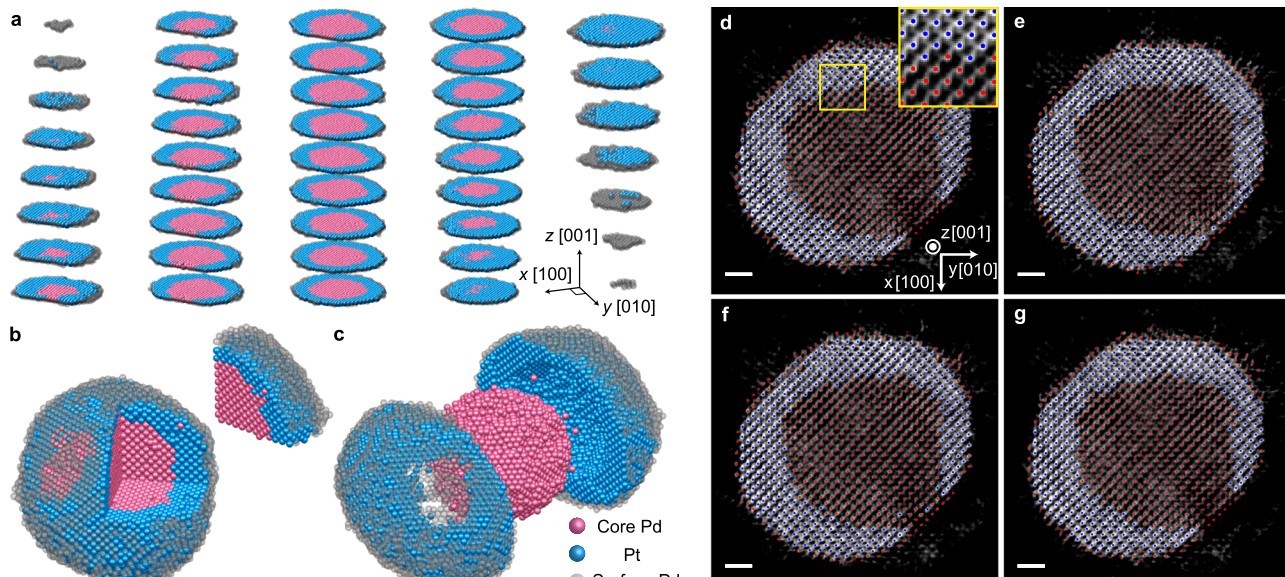

**Fig. 1 | Experimentally determined 3D atomic structure of a Pd@Pt core-shell nanoparticle. a** Atomic layer slices along the [001] crystallographic direction, to show the overall core-shell structure. **b** Overall 3D view of the nanoparticle with one octant of the sphere separated to visualize the core-shell architecture. **c** Overall 3D structure with separated core and shell. The shell is cut in half showing the core and shell interface structure. **d–g** 1.06 Å thick slices of the 3D tomogram, showing four consecutive atomic layers along the [001] direction, respectively. The grayscale background represents the intensity of the tomogram, and the red and blue dots represent the traced atomic coordinates of Pd and Pt, respectively. The Pd and Pt atoms can be well distinguished from the intensity contrast. The inset in **d** shows a magnified view of the area near the interface (indicated by a yellow box). Scale bar, 1 nm.

structure or lower dimensional (2D) imaging, which cannot provide the full 3D atomic structural detail of the core-shell interface[4,7,17,19,20]. Due to the intrinsic limitation of 2D projection-based analyses, most of the studies have only focused on cubic-shaped nanoparticles terminated by {100} facets[17,19–21], although {111} facets are technologically more important for their superior catalytic properties, for example, in the case of Pt-based catalysts[14,22,23]. Also, misfit strain not only affects the structure of the shell but can also propagate into the core. Nevertheless, the full 3D strain profiles of core-shell particles remain elusive due to the difficulty in analyzing the 3D internal structures. Moreover, the effect of interface strain on the surface atoms can include lattice expansion/contraction along the direction of the surface normal and the in-plane directions, related to the Poisson effect[21,24–26]. All spatial degrees of freedom must be considered for an accurate description of surface/interface atomic behavior such as a local atomic displacement and strain. Therefore, many studies emphasize the Poisson effect to properly understand nanomaterial structure-property relationships, but it has not been experimentally realized in 3D atomic detail[21,25–28].

Here, we reveal the largely unexplored nature of a 3D core-shell interface via atomic electron tomography (AET)[29–32] by using cuboctahedron-like core-shell nanoparticles with a Pd core and a Pt shell (Pd@Pt) as a model system, which has notable application potential especially in the field of catalysts[16,17,33–37]. We determined the full 3D arrangement of atoms for two such nanoparticles from the tomograms of atomic resolution. From the experimentally determined 3D atomic structures, full strain profiles of the core-shell nanoparticles were obtained, which revealed a direct correlation between the surface and interface strain as well as the extension of the strain field into both the core and shell regions. Strong shape-dependent anisotropy was observed from the strain distributions, whose origin was further investigated via molecular statics (MS) simulations. The opposite behavior of displacements along perpendicular directions, generally known as the Poisson effect, was clearly seen at the global nanoparticle scale and also on the scale of individual atomic bonds. Finally, based on the observed surface strains, oxygen reduction reaction (ORR) activities at the surface were predicted at the atomic scale via the strain-adsorption energy relation.

## Results and discussion

### Experimental identification of 3D atomic structure

Pd@Pt nanoparticles with a core-shell structure were synthesized by a previously reported method[16] (Methods). Using aberration-corrected microscopes operated in annular dark-field scanning transmission electron microscopy (ADF-STEM) mode, we measured tomographic tilt series for two nanoparticles: one with an average diameter of 8.56 nm (Particle 1) and another with an average diameter of 5.76 nm (Particle 2) (Methods). These nanoparticles are not fully spherical, and actual 3D shape and size information is provided in Supplementary Movies 3 and 7. After post-processing[29–31] of the obtained tilt series images, 3D tomograms of the two Pd@Pt nanoparticles (Particles 1 and 2) were reconstructed using the GENFIRE algorithm[38] (Methods). Full 3D atomic coordinates and chemical species of the atoms contained in each nanoparticle were determined from the tomograms by atom tracing and classification techniques with the precisions of 23.9 pm (Particle 1) and 24.6 pm (Particle 2) (Methods).

Figure 1a–c shows the determined 3D atomic model of Particle 1. The particle clearly exhibits the expected overall core-shell structure. Interestingly, the Pd core is not fully covered by the Pt shell, and a part of the Pd core is exposed to the surface near the (111) facet. The tomogram intensity and traced atom position maps given in Fig. 1d–g and Supplementary Movie 1 also confirm the observation, showing a clear core-shell structure with some part of the core Pd atoms being exposed.

While most Pd atoms are located in the core of the nanoparticle, some Pd atoms (lower intensity blobs in the tomogram; see Fig. 1d–g) were also observed on the particle surface. This is not surprising because residual Pd precursors in the growth solution were reduced on the surface of the generated core-shell nanoparticles during the synthesis. To verify the existence of surface Pd atoms, an energy-dispersive X-ray spectroscopy (EDS) experiment was conducted. The STEM-EDS map (Supplementary Fig. 1) shows the Pd signal near the surface, evidencing that the observed surface Pd atoms are real.

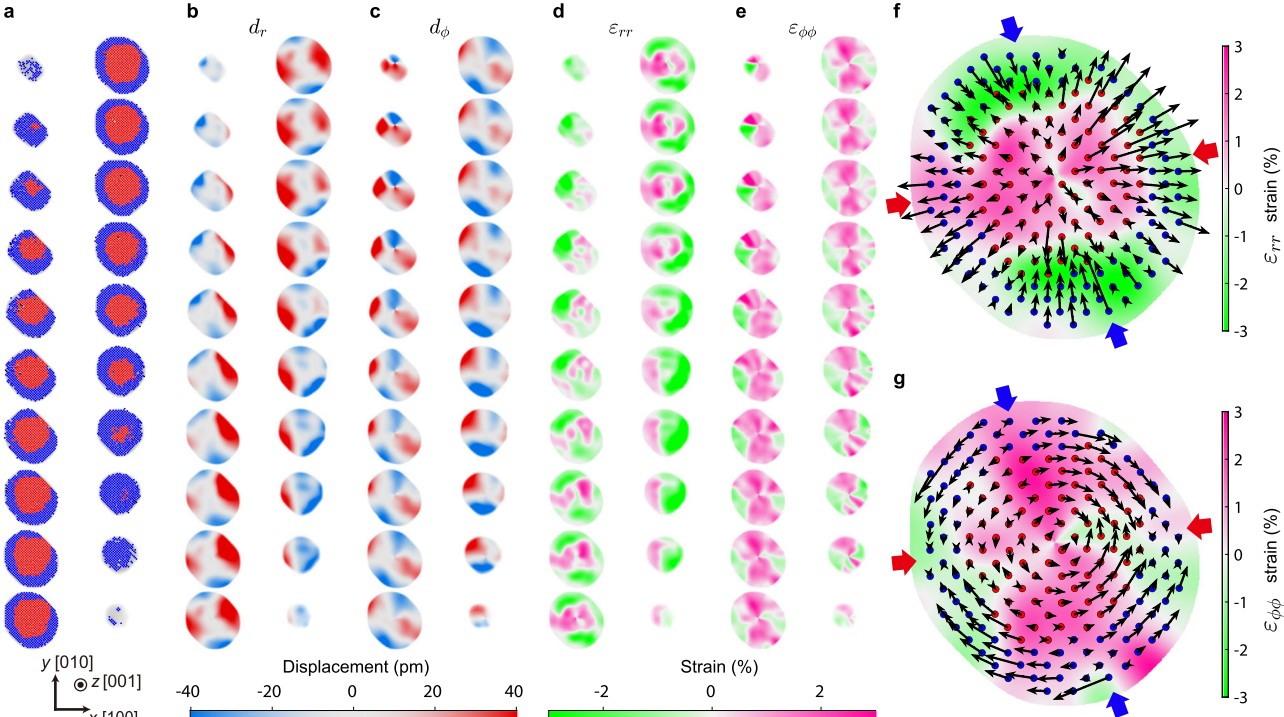

**Fig. 2 | 3D atomic displacements and strain maps of the Pd@Pt nanoparticle.**
**a** Atomic layers (sliced along the [001] direction) of the core-shell nanoparticle.
Note that only one layer per every two atomic layers is plotted. Red and blue dots
represent the positions of the Pd and Pt atoms assigned to the fcc lattice, respec-
tively, and black dots represent the positions of atoms not assigned to the fcc
lattice (Methods). **b-c** 3D atomic displacement maps, along the radial direction ($d_r$)
(**b**), and along the azimuthal direction ($d_\phi$) (**c**) in the spherical coordinate system.
**d-e** 3D strain maps in spherical coordinates, representing radial ($\varepsilon_{rr}$) (**d**) and azi-
muthal ($\varepsilon_{\phi\phi}$) (**e**) strains, respectively (Methods). The atomic displacement and

strain maps presented in **b-e** were calculated from the corresponding layer pre-
sented in **a**. **f** A map showing the 3D radial strain, atomic positions (only half of the
atomic positions are plotted for better visualization), and radial displacement
vectors of an atomic layer at the middle of the particle. **g** Similar plot with **f**, for
azimuthal strain and displacements. Note that the radial displacements ($d_r$) point
outward in the region where the azimuthal displacements ($d_\phi$) converge in (the
region pointed by red arrows), and the opposite behavior can be observed in the
region where radial displacements point inward, where the azimuthal displace-
ments diverge (the region pointed by blue arrows).

However, since our main interest lies in the relation between the
interfacial strain (induced from the lattice mismatch) and the Pt sur-
face structure, we focused on the core Pd atoms and shell Pt atoms for
further analysis.

Having full atomic structural information (3D atomic coordinates
and chemical species) allows us to quantitatively analyze the structural
properties of the nanoparticle. We first compared the 3D atomic
structure with an ideal face-centered cubic (fcc) lattice (Methods).
Most of the atoms can be assigned to fcc lattice sites, suggesting that
the core-shell nanoparticle can structurally be regarded as a single fcc
crystal, as can be seen in Supplementary Movie 2, Fig. 1b and the inset
of Fig. 1d. An fcc lattice was fitted to the atomic coordinates, and the
best-fit lattice constant was determined to be 3.90 Å. Local lattice
assignments and fittings were further performed for each atom in the
nanoparticle (Methods), resulting in the averaged local lattice con-
stants of 3.92 ± 0.06 Å for Pd and 3.87 ± 0.08 Å for Pt (the errors
represent the standard deviation). Interestingly, the averaged Pd lat-
tice constant shows a value larger than that of bulk Pd (3.88 Å)[39,40],
while the averaged Pt lattice constant is smaller than that of bulk Pt
(3.91 Å)[40,41], resulting in the Pd lattice constant exceeding that of Pt.
These findings are consistent with the reported shrinkage of the Pt
lattice when synthesized as nanoparticles[42,43] and the expected
expansion of the Pd lattice as a result of hydrogenation of the Pd
precursor in the synthesis process[44–48].

The averaged volumetric strains can be calculated from the local
lattice constants, which can be compared with those from a previous
X-ray absorption fine structure (XAFS) study of a similar Pd@Pt
system[49]. The averaged volumetric strains of Pd@Pt core-shell nano-
particles from the XAFS experiments are 1.6% for Pd and −0.3% for Pt.

Averaging the volumetric strain obtained from AET for every atom in
the two nanoparticles yields volumetric strains of 2.1 % for Pd and
−0.4% for Pt. Both of the results from XAFS and AET consistently show
relatively large tensile strains at the core part and mild compressive
strains in the shell. XAFS experiments sample a wide range of nano-
particle sizes to obtain the average strain information of the target
system to provide averaged information and are blind to any local
atomic-scale details. AET currently can only be applied to a few
nanoparticles rather than providing global information of nanoparticle
ensembles, but it can detect individual atom level details regarding the
core-shell system and provide detailed local information at the single-
atom scale throughout each individual nanoparticle.

**Displacement and strain analysis at the single-atom scale**
Since each atom was assigned to a specific site in an fcc lattice, the 3D
displacements between the atomic coordinates and corresponding fcc
lattice sites can be directly obtained. The displacements were mapped
onto a Cartesian grid by Gaussian kernel averaging, and 3D strain
tensor maps were obtained by taking their derivative (Methods).
Figure 2a–e shows the atomic structure, displacement map, and strain
map of Particle 1 for each atomic layer sliced along the [001] direction.
Again, a clear core-shell structure can be observed from the sliced
atomic structure map (Fig. 2a). The displacement maps (Fig. 2b, c)
show an interesting anisotropy throughout the nanoparticle, exhibit-
ing positive radial and azimuthal displacements along the x-direction
and negative displacements along the y-direction. Note that this
overall behavior of opposite displacement along the two perpendi-
cular directions is consistent with the Poisson effect for metallic
compounds[24–26]. In the radial strain map (Fig. 2d), the effect of the core-

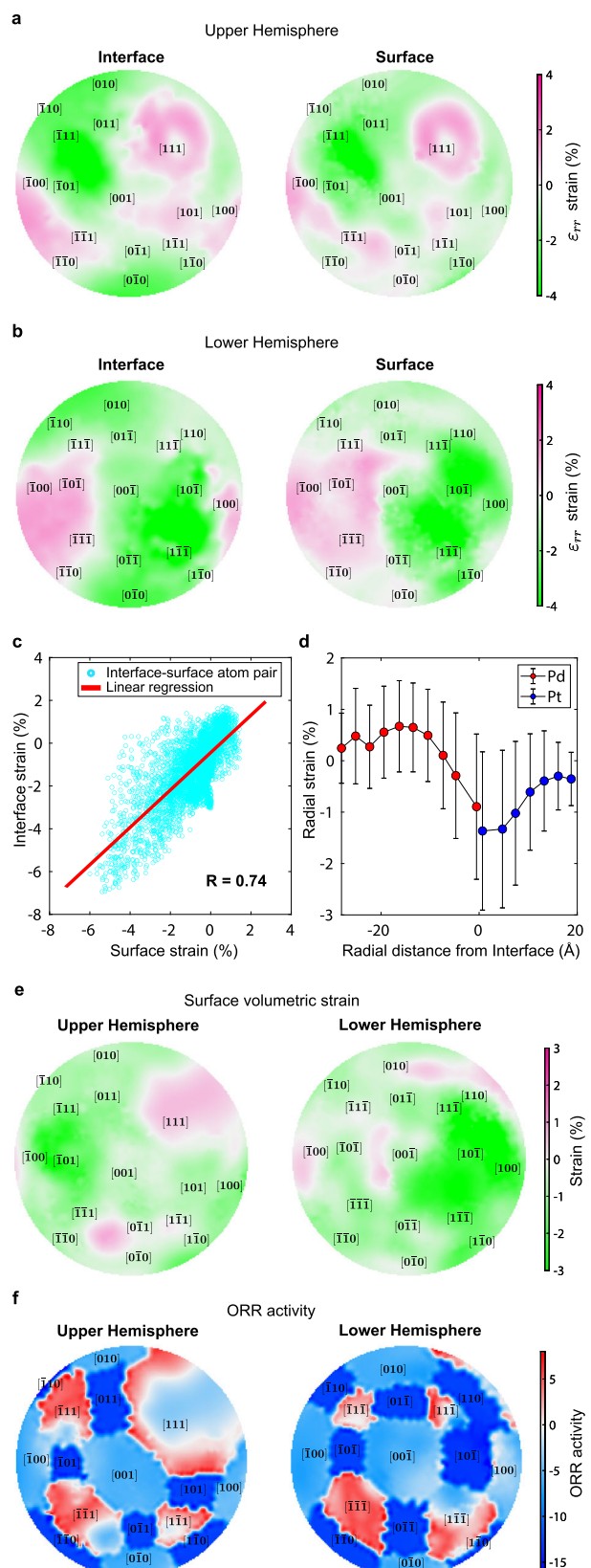

**Fig. 3 | Surface and interface strain behavior, radial strain profile, and surface ORR activity. a–b** Pole figures obtained from stereographic projections of the radial strain at the interface and surface for the upper (**a**), and lower (**b**) hemispheres, respectively. The Miller indices of low-index facets are marked at the average position of the surface atoms assigned to each facet (Methods). **c** A scatter plot showing the relation between the observed radial strains of surface and interface atoms. Each data point was obtained by paring each interface atom with the corresponding surface atom along the radial direction (Methods). A Pearson correlation coefficient (R) of 0.74 and slope of 0.87 were obtained from a linear regression (red line). **d** The radial strain plotted as a function of the distance from the interface. Each data point and error bar represent an average and standard deviation of the strain values within the bins of 3 Å distance interval, respectively. **e** Pole figures obtained by stereographic projections of the volumetric strain at the surface for the upper and lower hemispheres. **f** Pole figures obtained from stereographic projections of the surface ORR activity for the upper and lower hemispheres. The ORR activity is represented as $\ln (j/j_{Pt,(111)})$ where $j$ is the current density. The Miller indices of low-index facets are marked at the average position of the surface atoms assigned to each facet (Methods). Source data of **c** and **d** are provided as a Source Data file.

direction strain (Fig. 2e) does not show a clear bimodal distribution expected from the core-shell structure, but instead an alternating tensile and compressive behavior along the $\varphi$ direction can be seen, consistent with the anisotropy observed in the displacement map.

To analyze the local atomic-scale details of the core-shell structure, the atomic displacement vectors and strain maps for the atomic layer in the middle of the nanoparticle were investigated (Fig. 2f, g). The aforementioned Poisson effect is observed in the overall displacement along the $x$- and $y$- directions. Interestingly, we can also see local displacements at the atomic scale which resembles the global Poisson effect (i.e., opposite displacement along two perpendicular directions). In the region where radial displacements are most strongly pointing outward, the azimuthal displacements converge inward, i.e., the arrows meet head-to-head (the region pointed by red arrows in Fig. 2f, g). The opposite behavior is observed in the region of radial displacements pointing inward (converge), where the azimuthal displacements diverge outward, i.e., the arrows meet tail-to-tail (the region pointed by blue arrows in Fig. 2f, g). This not only emphasizes that atomic-scale tailoring of the local structural behavior is possible (controlling the in-plane displacement by tuning the radial strain, and vice versa), but also suggests that a simple unit cell volume-based description of surface strain[2,4,5,7,50] might not be sufficient to properly predict the surface catalytic activity, necessitating more detailed study incorporating the in-plane and out-of-plane strain behavior separately.

### Observing interface-surface correlation and strain profile

From the calculated strain information, we can directly address the relationship between the misfit strain at the interface and that at the surface, as visualized in Fig. 3a, b (pole figures) and Supplementary Movie 3 (Methods). It can be clearly seen from the pole figures and the 3D movie that the interface and surface strains overall exhibit very similar behavior, suggesting that they are strongly correlated. To quantitatively analyze these correlations, the radial strain values of each atom at the interface were paired with those of corresponding surface atoms as described in Fig. 3c (Methods). Again, a strong positive correlation between the interface and surface strains can be observed, and the Pearson correlation coefficient (R) of 0.74 and a slope of 0.87 were obtained via linear regression. To further verify the correlation, the out-of-plane strain was plotted following a path connecting high-symmetry facets (Supplementary Fig. 3 and Methods) for the interface and surface, separately. The results also clearly show that the interface and surface strains are directly correlated. The strong correlations we observe from several analyses described above indicate that the strain at the interface caused by the lattice mismatch can propagate through the thin shell and commensurately affect the

shell structure is clearly visible; tensile strains dominate in the core, while the shell mainly shows the compressive strain. The volumetric strain obtained by summing the three principal strain components in Cartesian coordinates (Supplementary Fig. 2k) shows a clear contrast between the core and shell regions confirming the existence of core-shell misfit strain throughout the nanoparticle. The azimuthal

surface strain, suggesting that the surface strain can indeed be controlled by turning the misfit strain at the interface.

The overall strain behavior of the nanoparticle along the radial direction was also investigated. Figure 3d shows the radial strain as a function of the distance from the interface (Methods). A weak tensile strain can be observed in the deep core region of the nanoparticle, and the strain gradually becomes compressive as the interface is approached. This behavior is consistent with the fact that the Pd core has a larger local lattice parameter than that of Pt. In the Pt shell region, the strain gradually increases from the interface toward the surface, showing the expected lattice relaxation behavior[4,7,17]. Note that the propagation of misfit strain towards the center of the core is also prominent, not only towards the surface of the shell. This suggests that the misfit strain effect for both core and shell should be considered together for fine tailoring of the core-shell structure, for which the determination of the full 3D strain profiles will be essential.

## Calculation of the ORR activity from surface strain

Having a full 3D atomic structure allows a direct calculation of the surface ORR activity (Fig. 3f) via the relation between the surface volumetric strain (Fig. 3e), type of facet, and OH binding energy[14,22,51,52] (Methods). A clear facet-dependence can be observed from the ORR activity (Fig. 3f and Supplementary Movie 4), showing much higher ORR activity for {111} facets compared to that of {100} and {110} facets, as expected from the known big difference between the ORR activity of different facets[14,22,23] (Supplementary Fig. 4). While the ORR activity does depend primarily on the surface facets, we also observe a strong dependence of activity on the local strain. For the {111} facets, the ORR activity becomes low in the regions of tensile strain, and high ORR activity is observed at compressively strained parts (Fig. 3e, f and Supplementary Movie 4). Interestingly, due to the interface-induced strain, most of the {111} facets show enhanced ORR activity compared to unstrained cases, suggesting that the ORR activity can indeed be controlled by the proper design of the surface strain, as reported in a recent study[19].

Note that the strain maps show an obviously anisotropic distribution for both the interface and the surface. Surprisingly, there is no strong relation visible between the observed anisotropy and identified facet types ({100}, {110} or {111}). The strain curve connecting high-symmetry facets (Supplementary Fig. 3) further confirms this behavior; the type of facets is not directly related to the anisotropy. Considering the fact that the region of the exposed Pd core shows a particularly strong tensile strain, we speculate that the anisotropy is mainly due to the core-shell geometry and overall shape rather than the facet structure.

## MS simulation of core-shell structures

To verify the interface-surface correlation and investigate the origin of the strain anisotropy, MS simulations were performed using the same core-shell geometry and shape of the experimentally measured nanoparticles. Starting from an ideal fcc lattice model fitted to the experimental atomic structure of Particle 1, the atomic coordinate relaxation was simulated at 0 K using the embedded atom model (EAM) potential (Methods). Note that we used the chemical species of Ag and Al instead of Pd and Pt (Methods), to mimic the lattice constant ratio between hydrogenated Pd and Pt while capturing the elastic anisotropy of cubic crystals. Although several EAM potentials[53,54] for the binary Pd-Pt system exist, they cannot describe the ternary interaction involving hydrogen and cannot account for the dilatation of the Pd lattice constant from hydrogenation. Because of the elastic isotropy originated from the Cauchy constraint, $C_{12} = C_{44}$, a pair potential such as Lennard-Jones (LJ) model is not suitable to describe the strain distribution inside the nanoparticle, although we can easily tune the parameters to exactly match the ratios of lattice constants and bulk moduli. As shown in Fig. 4a–c, the simulation reproduces several features observed in the experiment: positively strained core, compressively strained shell, and the high tensile strain of the region where the core is exposed to the surface. Alternating compressive and tensile behavior of the $\varepsilon_{\varphi\varphi}$ strain along the azimuthal direction is also visible. The pole figures for the interface and surface radial strains (Fig. 4d, e) indicate a strong correlation between the surface and interface strain. MS simulations performed with a truncated LJ potential designed to match the ratios of lattice constants and bulk moduli show a notably distinct strain distribution (Supplementary Fig. 15), which manifests that the elastic anisotropy of cubic crystals plays a crucial role to induce the experimentally observed strain distribution and simple explanation based on Poisson effect is insufficient to explain it.

The strong anisotropy of the radial strain is also well reproduced, supporting our speculation that the core-shell geometry and shape are the main parameters governing the anisotropy. To further verify this, we conducted two additional simulations (Supplementary Figs. 5 and 6). We first tried a similar MS relaxation using an atomic model of a perfectly spherical core-shell nanoparticle. As can be seen in Supplementary Figs. 5b and 6b, no clear anisotropy can be found from this simulation, showing a symmetric distribution of strains. In contrast, if we change the initial shape of the nanoparticle by partially cutting the spherical core-shell to expose a part of the core to the surface, the simulation produces more anisotropic strain behavior similar to the experiment (Supplementary Figs. 5c, 6c, and 7). These results demonstrate that the anisotropic distribution of strains observed in our experiment is mainly governed by the core-shell geometry and shape of the nanoparticle, rather than the facet structure.

## The behavior of the Particle 2

All the analyses (displacement, strain, and ORR calculation) were conducted in the same way for Particle 2 (Supplementary Figs. 8–14 and Supplementary Movies 5–8). There are a few different features found compared to those of Particle 1. First, the displacements and strains are distributed differently (Supplementary Figs. 9 and 11). The difference may originate from a combination of effects; the two particles have different core sizes, shell thickness, and overall shape (especially, there is no exposed Pd core on the surface in the case of Particle 2). Second, the surface-interface strain correlation is weaker for Particle 2 compared to that of Particle 1. It can be attributed to the larger surface atomic deviation from the lattice positions for Particle 2, which can result from the relatively larger surface curvature (i.e., smaller radius). Particle 2 shows a larger root-mean-square deviation (RMSD) between the surface atoms and their lattice positions compared to that of Particle 1 (68.5 pm and 71.2 pm RMSDs for Particle 1 and Particle 2, respectively), supporting our assertion. Third, the enhancement of ORR activities at the {111} facets is less pronounced (Fig. 3f and Supplementary Fig. 11f: ORR pole figures). This is related to the distributions of the surface volumetric strains (Supplementary Figs. 11e, f, 13 and Supplementary Movie 8); the peak of the surface strain distribution for Particle 2 shows slightly more tensile behavior compared to that of Particle 1 (consistent with the larger local lattice constant of the Pt shell of Particle 2), resulting in relatively poorer ORR activity. This can be explained by the thickness effect: the Pt surface lattice is not fully relaxed due to a thinner Pt shell thickness (Supplementary Figs. 11d and 25). From Supplementary Fig. 20, it can be seen that the in-plane lattice constant gradually decreases as the distance from the interface increases for both Particle 1 and Particle 2, showing a clear thickness-dependent lattice relaxation behavior. Considering the average shell thickness of 9.75 Å for Particle 1 and 6.77 Å for Particle 2 (Methods and Supplementary Fig. 25), this analysis demonstrates that the relatively larger local surface lattice constants of the Pt shell observed for Particle 2 can be (at least partially) attributed to the thickness effect; since the lattice constant decreases as a function of the distance from the interface, the surface lattice constant is expected to be smaller for a thicker shell. However, as can be seen from the

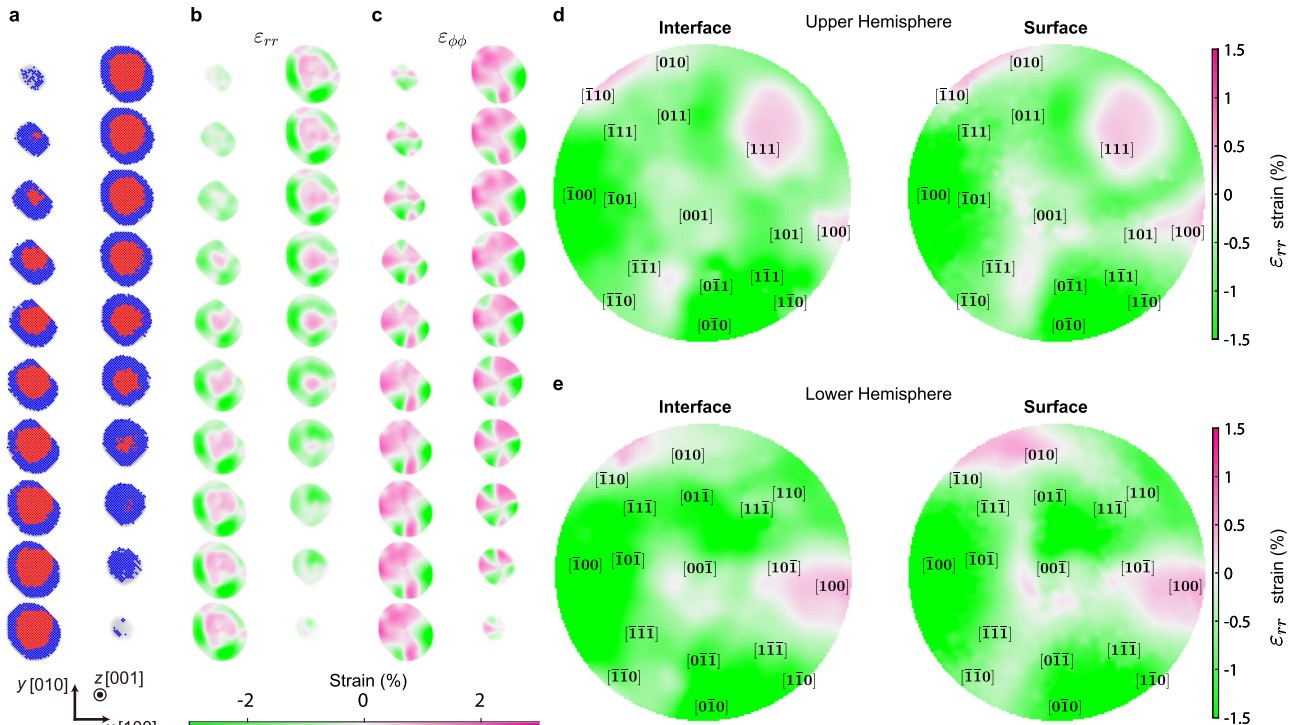

**Fig. 4 | 3D strain maps and pole figures of the core-shell structure obtained by an MS simulation. a** The core-shell atomic structure resulting from the MS simulation (Methods), which is divided into atomic layers along the [001] direction. Note that only one layer per every two atomic layers is plotted. The red and blue dots represent the positions of Pd and Pt atoms assigned to the fcc lattice, respectively. **b–c** The calculated strain maps from the MS simulation result for the radial ($\varepsilon_{rr}$) (**b**) and azimuthal ($\varepsilon_{\phi\phi}$) (**c**) strains in the spherical coordinate system, respectively. The strain maps were calculated from the corresponding layers presented in **a**. **d–e** Pole figures showing the interface and surface strains of the MS simulation result. The pole figures were obtained from stereographic projections of the radial strain at the interface and surface for the upper (**d**), and lower (**e**) hemispheres, respectively. The Miller indices of low-index facets are marked at the average position of the surface atoms assigned to each facet (Methods).

differences in the interface lattice constants and lattice relaxation rates, other effects (the overall shape, size, larger core lattice constant [this might be due to the different degree of hydrogenation of the Pd], etc.) can also affect the behavior of the surface lattice constant. Experimental ORR activity measurements were also performed on two groups of nanoparticles with similar sizes to Particle 1 and Particle 2 (Methods and Supplementary Figs. 22 and 23). From Supplementary Fig. 23, it can be seen that the group of particles having a size similar to Particle 1 shows a better ORR activity than the group with the size similar to Particle 2.

In spite of these differences, the overall core-shell effects we discussed above can be consistently observed in the two nanoparticles. The overall strain distribution shows tensile and compressive radial strains in the core and shell, respectively, while azimuthal strains exhibit alternating behavior (Supplementary Fig. 9d, e). The Poisson effect can also be observed at the global and local scales (Supplementary Fig. 9f, g). Strong surface-interface correlations (Supplementary Figs. 11a–c, 12 and Supplementary Movie 7) and shape-dependent strain anisotropy (Supplementary Figs. 11a–b, 12, 14 and Supplementary Movie 7) can be clearly seen as well. The facet-dependent ORR distribution with a clear strain-induced enhancement is also found (Supplementary Fig. 11e, f and Supplementary Movie 8). This suggests that the overall structural behaviors (surface-interface correlation, Poisson effect, shape-dependent anisotropic strain, facet/strain dependent ORR activities) represent a general core-shell structural effect.

In summary, the full 3D atomic structures of core-shell nanoparticles were determined via AET, and their 3D displacement and strain distributions were mapped, leading to the following conclusions. First, partial exposure of the core part on the surface resulted in a remarkable tensile strain effect in the local region for the larger nanoparticle. Second, the misfit of lattice and thus intrinsic strain can be observed both on the surface and along the interface, and the relationship between them suggests a linear tendency. Third, the surface displacement behavior clearly demonstrated the Poisson effect globally as well as locally. Fourth, the prominent anisotropy of strain displacement shall be attributed to the core-shell geometry and the shape of the nanoparticles, rather than surface facet effects, as confirmed by MS simulations. These observations suggest that the strain at the surface can be controlled through the misfit strain at the interface as well as core-shell design, which would allow the optimization of catalytic activity as desired. These characterization results reveal the 3D structure of the core-shell nanoparticles at the individual atom level, directly providing the structure-property relations regarding strain-engineerable core-shell systems. Looking forward, controlled AET experiments via fine-tuning of the particle shape, core size, shell thickness, and the degree of hydrogenation of the Pd can shed light on the size and thickness effect, which can bring valuable information in the field of nanocrystal fabrication and catalysis. Furthermore, if in-situ capability can be properly implemented in the AET by overcoming some technical obstacles (e.g., sample durability, time resolution, and TEM holder design), it will allow us to unveil the 3D atomic dynamics of nanomaterials during dynamical processes such as segregation/diffusion, sintering, and device operation.

## Methods
### Synthesis of Pd@Pt nanoparticles
Pd@Pt nanoparticles were synthesized by using a reported method[16]. 50 mg of Na$_2$PdCl$_4$ and 94.4 mg of polyvinylpyrrolidone were dissolved in 10 mL of ethylene glycol with vigorous stirring under an Ar

atmosphere. Then, the solution was heated to 200 °C with a ramping rate of 4.375 °C min$^{-1}$ and incubated at 200 °C for 1 h. In this step, Pd nanoparticles were formed. The resultant solution was cooled down to room temperature for the Pt shell growth process. For the formation of the Pt shell, the solution was heated to 160 °C with a ramping rate of 1.5 °C min$^{-1}$. When the temperature reached 110 °C, 0.53 mL of an aqueous solution of $K_2PtCl_4$ (160.6 mM) was injected into the solution. After incubating the solution at 160 °C for 3 h, it was cooled to room temperature. Acetone and hexane were added to the reaction batch, and the resultant Pd@Pt nanoparticle colloidal solution was stored and used for further characterizations.

### Sample preparation for AET characterization

The initially prepared Pd@Pt nanoparticles were collected by centrifugation (1743 rcf, 30 min) and washed with a mixture of ethanol and acetone, and washed again twice with pure ethanol. The washed nanoparticles were collected again and dispersed in an aqueous solution containing 100 mM cetyltrimethylammonium chloride (CTAC) to prevent the aggregation of nanoparticles during the sampling procedure. In this step, the concentration of nanoparticles was diluted to a concentration 2 or 3 times lower than that of the original colloidal solution. The CTAC-treated nanoparticles were washed with water and re-dispersed in water. This solution was used to prepare the specimens for electron tomography.

### Data acquisition

Specimens for electron tomography measurements were prepared by depositing a solution of the synthesized Pd@Pt nanoparticles dispersed in water onto a 5 nm thick silicon nitride membrane grid (Particle 1), and a 3–4 nm thick carbon membrane grid (Particle 2). After deposition, the grids were annealed in a vacuum at 150 °C (Particle 1), and 100 °C (Particle 2) for 24 h. Two tomographic tilt-series were acquired from the Pd@Pt nanoparticles using an FEI Titan Themis3 double Cs corrected and monochromated transmission electron microscope (TEM) at the Korea Basic Science Institute in Seoul (Particle 1) and the double Cs corrected TEAM 0.5 microscope equipped with TEAM stage at the National Center for Electron Microscopy in the Lawrence Berkeley National Laboratory (Particle 2), respectively. The images were acquired using annular dark-field (ADF) scanning transmission electron microscopy (STEM) mode. For Particle 1, we collected a tilt series at 33 different tilt angles in the angular range of −68.2° to +68.2° using 300 kV acceleration voltage (Supplementary Fig. 16), detector inner and outer semi-angles of 38 mrad and 200 mrad, respectively, a convergence beam semi-angle of 25.2 mrad, and a beam current of 10 pA. For Particle 2, we collected a tilt series at 42 different tilt angles in the angular range of −66.1° to +61.8° using 200 kV acceleration voltage (Supplementary Fig. 17), detector inner and outer semi-angles of 43 mrad and 216 mrad, respectively, a convergence beam semi-angle of 30 mrad, and a beam current of 18 pA. For both particles, four consecutive 1024 × 1024 pixels images were acquired for each tilt angle with 3 μs dwell time with a pixel size of 0.353 Å for Particle 1 and 0.329 Å for Particle 2, respectively. Each tilt series measurement took about 4 h. To check if the nanoparticles have suffered from the beam-induced structural change during the tilt series acquisition, the zero-degree projections were measured three times during the acquisition (at the beginning, in the middle, and at the end of the experiment). The zero-degree images show that the nanoparticles were slightly rotated during the experiment. We obtained the rotation angles by comparing the experimental zero-degree images with the forward-projection images of 3D atomic structure (see below for the atomic structure determination process) along several different tilt angles near zero-degree and finding the best-matching projection angles. Each zero-degree projection is consistent with the best-matching forward-projection image, evidencing that the atomic structure did not suffer from substantial structural change during the experiment (Supplementary Figs. 18, 19).

### Image post-processing, GENFIRE reconstruction, and tomogram post-processing

Image post-processing (drift correction, scan distortion correction, image de-noising through BM3D noise reduction[55,56], tilt-series alignment based on center-of-mass and common line method) was conducted for each tilt series as described in previous works[29–31]. After the post-processing, 3D tomograms were obtained from the aligned tilt-series images using the GENFIRE reconstruction algorithm[38]. To improve the tomogram quality and reduce the tilt angle error, GENFIRE-based spatial re-alignment was also conducted[31,38]. Final 3D tomograms were obtained by running GENFIRE with the following parameters: discrete Fourier transform interpolation method, number of iterations 1000, oversampling ratio 4, and interpolation radius 0.1 pixel. It usually takes about 2-3 days to go through these processes.

### 3D identification of atomic coordinates and chemical species

The 3D atomic coordinates of all atoms in each nanoparticle were determined by the atom tracing procedure[30,31] as follows. All 3D local maxima positions of the tomogram were found and sorted based on the peak intensity in descending order. Starting from the highest intensity local maxima, a volume of 5 × 5 × 5 voxels centered on the local maximum position was cropped, and a 3D Gaussian function was fitted for the intensity volume. If the fitted position does not violate the minimum distance constraint of 2.0 Å compared to any of the fitted positions in the traced atom list, the newly fitted position is added to the traced atom list. However, owing to the intensity elongation effect of the tomogram along the missing-wedge direction and slight imperfectness of the reconstruction, several connected intensity blobs can be found in the tomogram, for which the local maxima cannot be reasonably determined. To find the proper atomic positions of these intensity blobs, two additional atom tracing procedures were conducted as follows.

(a) The tomogram was sliced along the fcc [001] direction for each atomic layer, and the 2D local maxima in each slice were identified. Next, the same 3D Gaussian fitting procedure was repeated using the determined 2D local maxima positions from all the atomic layers with various cropped volume sizes (side lengths of 3–7 pixels). The fitted position from the volume size which showed the smallest residual of the fitting was added to the traced atom list if it does not violate the minimum distance constraint of 2.0 Å (Particle 1) or 1.7 Å (Particle 2).

(b) For each atomic layer slice, a 2D atom number density was calculated for each pixel by counting the number of atoms within the distance of 1.9 Å from each pixel. We identified the connected regions where the 2D atom number density becomes 0 and classified them into three different categories: region of artifacts due to imperfect tomogram reconstruction (for the regions whose areas are below 3.03 Å$^2$), region of one-atom candidates (for the regions whose areas are between 3.03 Å$^2$ and 5.63 Å$^2$), and region of two-atom candidates (for the regions whose areas are above 5.63 Å$^2$). The criteria for classifying these three cases were determined by estimating the range of area of connected zero density regions from an fcc atomic model with random spatial displacement which gives 22 pm root-mean-square deviation (RMSD) from the ideal fcc lattice sites. From the fcc model, we randomly removed one or two adjacent atoms and obtained the minimum area (i.e., the classification criteria) of the zero density region by calculating the atom number density as mentioned above. For each one-atom candidate region, an initial fitting point was obtained as the center of each region. For each two-atom candidate region, two initial fitting points were obtained from the Expectation-Maximization algorithm for the Gaussian mixture model[57]. Starting from those initial fitting points, the same Gaussian fitting method with varying volume sizes was applied.

After this process, 3D atomic models of 22,494 (Particle 1) and 7,143 (Particle 2) atoms were obtained. To finalize the 3D atomic structure, a manual correction was applied to add (or remove) physically (un)reasonable atom candidates. A minimum distance constraint of 1.7 Å was used during this process. Total 749 (Particle 1) and 489 (Particle 2) atoms were manually added, and 899 (Particle 1) and 771 (Particle 2) atoms were manually removed, resulting in the final 3D atomic models of 22,344 (Particle 1) and 6,861 (Particle 2) atoms. It takes about 1 day to run the automated atomic coordinate identification procedures, and the manual correction usually takes about 1–2 days.

After the manual correction of atomic positions, their chemical species (either Pt or Pd) were determined. To classify them, we used an unbiased atom classification method based on the k-means clustering algorithm[30] as follows. (a) We calculated the integrated intensity for every atom by summing all values within a box of $3 \times 3 \times 3$ voxels centered on each rounded atom position from the 3D tomogram. Histograms of the integrated intensities for all atoms are shown in Supplementary Fig. 26a for Particle 1 and Supplementary Fig. 26d for Particle 2. Next, we set the initial threshold as the average value of the integrated intensities to initially separate the Pt (the atoms with the integrated intensity higher than the threshold) and Pd (the atoms with intensities lower than the threshold). From the initial Pt and Pd atoms, we defined the averaged intensity box of $5 \times 5 \times 5$ voxels for each species by averaging over the $5 \times 5 \times 5$ voxels centered on all atoms of the same chemical species from the 3D tomogram. (b) For each identified atom, two error functions were calculated,

$$E_{Pt} = \sum_i |P_i - A_i^{Pt}|, \quad E_{Pd} = \sum_i |P_i - A_i^{Pd}| \qquad (1)$$

where $P_i$ is the $i$-th voxel intensity, and $A_i^{Pt}$ and $A_i^{Pd}$ are the $i$-th voxel intensity of the averaged intensity box for Pt and Pd, respectively. Using the error functions, all the atoms were re-classified into Pt or Pd based on the minimal error function. (c) From the updated atomic species classification, we re-calculated the averaged intensity boxes for Pt and Pd and the resulting error functions to classify all the atoms again. This step was repeated until there was no change in the species classification, resulting in 11,474 Pt and 10,870 Pd atoms for Particle 1 (Supplementary Fig. 26b, c), and 3435 Pt and 3426 Pd atoms for Particle 2 (Supplementary Fig. 26e, f).

## Assignment of experimental atomic coordinates to ideal fcc lattice sites

The atomic coordinates determined from the 3D tomograms were assigned to ideal fcc lattice sites using the following procedure. Note that since our main interest lies in the relation between the interfacial strain and the Pt shell surface structure, the surface Pd atoms were not included in the lattice assignment analysis. First, an atom closest to the mean position of the given 3D atomic coordinates was assigned to the origin of an fcc lattice. Then, the nearest neighbor positions of the atom were calculated based on initial fcc lattice vectors of lattice constant 3.88 Å (the lattice constant of bulk Pd[39,40]). For each nearest neighbor position, if there was an atom within 25% of its nearest neighbor distance (initially, $\frac{3.88 \text{Å}}{\sqrt{2}} \times 25\% = 0.686$ Å), the atom was added to the corresponding fcc lattice site. The nearest neighbor search was repeated for all the newly assigned fcc lattice sites. The process was repeated until no more atoms could be assigned to the lattice. Second, new fcc lattice vectors were fitted (fitting parameters: translation, 3D rotation, and lattice constant) to the atoms assigned to the lattice by minimizing the error between the measured atomic positions and the corresponding lattice positions of the fitted fcc lattice. These two steps were continuously repeated with the newly obtained fcc lattice vectors until there is no change in the fitted lattice vectors. After this process, 99.7% (Particle 1) and 97.1% (Particle 2) of the target atoms were

successfully assigned to the fcc lattices, and RMSDs between the assigned atom positions and the fitted fcc lattices were 49.83 pm and 58.95 pm for Particle 1 and Particle 2, respectively. The fitted fcc lattice constants were 3.90 Å (Particle 1) and 3.98 Å (Particle 2).

## Local lattice constant calculation

For local property analysis, a local lattice constant for each atom was calculated using the following procedure. First, for each atom in the 3D atomic model, we prepared a set of atoms that only contains atoms that correspond to the nearest neighbor sites of the given atom, based on the globally fitted lattice. Second, by setting the given atom position as the origin, the nearest neighbor fcc lattice sites of the given atom were calculated. In this calculation, the lattice constant of initial fcc lattice vectors were set as the lattice constant of the globally fitted fcc (3.90 Å for Particle 1 and 3.98 Å for Particle 2). For each nearest neighbor position, if there was an atom within the 25% of the nearest neighbor distance of the globally fitted lattice (0.690 Å for Particle 1, and 0.703 Å for Particle 2), the atom was assigned to the local fcc lattice site corresponding to the given nearest neighbor site. Note that we only considered the atoms in the nearest neighbor atom set defined above in this process. This assignment process was repeated for all the calculated nearest neighbor sites. The lattice of these assigned atoms was defined as the local lattice. Third, new fcc lattice vectors were fitted (fitting parameters: translation, 3D rotation, and lattice constant) to the atoms assigned to the local lattice by minimizing the error between the measured atomic positions and the corresponding lattice positions of the fitted local fcc lattice. Fourth, the second and third steps were continuously repeated with the newly obtained fcc lattice vectors until there is no change in the fitted lattice vectors. All these four steps were performed for all the atoms which were assigned to the globally fitted lattice. For each atom, the local lattice constant was determined from the obtained local fcc lattice. The averaged local lattice constants for each species were $3.92 \pm 0.06$ Å (Particle 1) and $4.00 \pm 0.11$ Å (Particle 2) for Pd, and $3.87 \pm 0.08$ Å (Particle 1) and $3.91 \pm 0.12$ Å (Particle 2) for Pt, respectively (the errors represent the standard deviation).

## Precision estimation for the atomic coordinates using multislice simulation

To estimate the precision of the measured atomic coordinates, we performed precision analyses[29,30] using multislice simulations[58–60]. From the experimentally measured atomic coordinates, a total of 33 (Particle 1) and 42 (Particle 2) projections, which correspond to the experimental tilt angles, were obtained using multislice simulations. The multislice simulations were performed under the same conditions with the experimental microscope parameters: 300 keV electron energy, 130 nm $C_3$ aberration, 0 mm $C_5$ aberration, 25.2 mrad convergence semi-angle, and 38 mrad and 200 mrad detector inner and outer semi-angles for Particle 1, and 200 keV electron energy, 0 nm $C_3$ aberration, 5 mm $C_5$ aberration, 30 mrad convergence semi-angle, and 43 mrad and 216 mrad detector inner and outer semi-angles for Particle 2. We also applied 16 frozen phonon configurations and 2 Å slice thickness for the simulations. 3D tomograms were obtained from the multislice simulated tilt series using the GENFIRE algorithm. The atomic coordinates identification process was performed for the 3D tomograms. By comparing the experimental atomic coordinates and the atomic coordinates traced from the multislice simulated tomograms, common atom pairs were identified; if the distance between the experimental atomic coordinates and multislice simulated atomic coordinates was less than a threshold distance (half of the nearest-neighbor distance of the fitted ideal fcc lattice), the pair of atoms was assigned as a common atom pair. RMSDs between all the determined common atom pairs were 23.9 pm (Particle 1) and 24.6 pm (Particle 2), respectively.

## Displacement field and strain calculation

Since each atom was assigned to a specific site in the fcc lattice, the atomic displacement vectors between the atomic coordinates and corresponding fcc lattice sites can be directly obtained in Cartesian coordinates. The displacements were mapped onto Cartesian grids by kernel averaging with optimized Gaussian kernels[29] (σ = 4.67 Å for Particle 1 and σ = 4.31 Å for Particle 2). 3D strain tensor maps were obtained by taking derivatives of the displacements. Using the transformation matrix between the Cartesian coordinate system and the spherical coordinate system, a vector transformation was applied to the atomic displacement vectors at each Cartesian grid point to obtain the radial ($d_r$) and azimuthal ($d_\phi$) components of displacement vectors in the spherical coordinate system. Similarly, using the same transformation matrix, a second-order tensor transformation was applied to the strain tensor of each grid point to obtain the radial ($\varepsilon_{rr}$) and azimuthal ($\varepsilon_{\phi\phi}$) principal strain components in the spherical coordinate system. Note that the strain tensor components at the positions of each atom (rather than Cartesian grid points) were also obtained by kernel averaging as illustrated in Supplementary Movies 3 and 7. We calculated the radial and azimuthal principal strains along all the possible three orthogonal zenith directions and the results were essentially identical under the claimed strain properties. For generating pole figures of the interface and surface strain, the surface/interface atomic models which only contain the surface/interface atoms of the full 3D atomic model were cut in half along the [001] direction. For each half, a stereographic projection of the atomic positions was obtained on a 2D plane[61]. Then, the radial strain assigned to each atom was interpolated and/or extrapolated on a 2D planar grid at 1 Å pixel$^{-1}$ spacing using the biharmonic spline method[62,63].

## Gaussian kernel parameter optimization for kernel averaging

The Gaussian kernel standard deviation (σ) parameters were optimized with the following procedure. First, an atomic model which consists of the atomic coordinates of the fitted ideal fcc lattice was prepared. Then, a radial strain function was generated based on the Fourier series (maximum frequency 0.21 nm$^{-1}$) with randomly chosen coefficients, and the atomic coordinates of the atomic model were adjusted to reflect the generated strain. The atomic coordinates were further randomly modified so that the atomic structures before and after applying the random modification have the same RMSD value as the one obtained from the precision estimation step. Using this atomic model, the strain calculation process was performed with several different Gaussian kernel standard deviations. The standard deviation value which showed the minimum error between the applied strain and calculated strain was chosen as the optimized kernel parameter. This process was repeated 1000 times with different randomly generated radial strains. The 1000 optimized kernel parameters were averaged to obtain the final Gaussian kernel standard deviation parameters, which are σ = 4.67 Å for Particle 1 and σ = 4.31 Å for Particle 2, respectively.

## Determination of the interface and surface atom pairs

To analyze the strain relationship between the interface and surface, we paired the surface atoms and interface atoms along the same radial directions. First, the surface atoms were determined by applying the alpha shape algorithm[64] to the 3D atomic models with shrink factor 1, and the initial interface atoms were determined by applying the same algorithm to the core Pd atoms. Interface Pt atoms were defined as the Pt atoms which are the nearest neighbors of any of the initial interface atoms, and the interface Pd atoms were defined as the Pd atoms which are the nearest neighbors of any of the interface Pt atoms. The union of interface Pt and interface Pd atoms were defined as the final interface atoms. Some atoms were categorized as both surface and interface atoms (the Pd atoms on the exposed surface of Particle 1), which were excluded from the atom pairing analysis. Next, the position vectors of

all interface atoms were calculated, for which the origin was set as the averaged 3D coordinates of all the core Pd atoms. For each position vector, an extension line was drawn along the direction of the position vector. Then, the surface atom which has the smallest distance to the line was paired with the corresponding interface atom represented with the position vector. This process was applied to all the interface atoms, and the distance between the interface atom and the paired surface atom is defined as the shell thickness for each interface atom.

## Definition of the distance from the interface for each atom

For each atom, the distance from the interface along the radial direction was determined as follows. First, the position vectors of all the atoms were calculated, for which the origin was set as the averaged 3D coordinates of all the core Pd atoms. For each position vector, an extension line was drawn along the direction of the position vector. Next, the interface atom which has the smallest distance to the line was paired with the given atom represented by the position vector. Then, the distance between the given atom and the paired interface atom was defined as the distance from the interface. This process was applied to all the atoms in the 3D atomic models.

## Strain calculation along the paths connecting the high-symmetry facets

To verify the strain correlation between the surface and interface, the out-of-plane strains were calculated following different paths connecting high-symmetry facets for the surface and interface separately. Here, we employed two different types of paths, one connecting through ({100}, {110}, {111}) facets and another passing through {100}, {110} facets (Supplementary Fig. 3a, b). The facets were defined as the outermost atomic planes which contain more than the given threshold number of atoms, perpendicular to the corresponding facet directions for the core Pd atoms (for interface facets) and shell Pt atoms (for surface facets). If the number of atoms in the outermost atomic plane is less than or equal to the given threshold number, the next outermost atomic planes are added to defined facets until the number of atoms in the facet is more than the given threshold number or the number of atomic planes in the facet is equal to three layers. Note that the threshold numbers for the outmost plane were chosen as 10 and 5 atoms (Particle 1), and 4 and 2 atoms (Particle 2), for the core and shell, respectively. For each facet atom, the out-of-plane strain was calculated by taking derivatives of the out-of-plane displacements which were obtained by kernel averaging with optimized Gaussian kernels (σ = 4.67 Å for Particle 1 and σ = 4.31 Å for Particle 2). The strain of each facet was defined as the average of the out-of-plane strains for all the atoms in the facet.

## ORR activity calculation at the surface

Since the surface ORR activity strongly depends on the type of facets as well as the surface strain, we classified all surface atoms into facets of the three dominant facet families: {111}, {100} and {110}. Most of the surface atoms were successfully assigned to one of the three dominant facet families by applying the method described above with the threshold number of 12 (the number of atomic planes in the facet can be more than three layers in this case). For the surface atoms which are not assigned to any of those facets, we calculated the distance from the surface atom to the deepest atomic layer of each facet ({111}, {100} or {110}) by measuring the number of atomic layers between them. The surface atom is then assigned to the facet with the smallest distance. If two or more facets have the same smallest distance from the surface atom, we calculated the distance from the surface atom to all the atoms already assigned to one of the dominant facet families, and the surface atom was classified into the facet which contains the atom of the smallest distance. Therefore, all surface atoms are assigned to one of the three dominant facet families (unlike our previous approach[50] in which some surface atoms are excluded from ORR analysis). Next, the

local volumetric strain for each surface atom was calculated as $(a_{local} - a_{ref})/a_{ref}$, where the $a_{ref}$ is the bulk Pt lattice constant[41] 3.91 Å and $a_{local}$ is the kernel averaged local lattice constant with the optimized Gaussian kernels ($\sigma$ = 4.67 Å for Particle 1 and $\sigma$ = 4.31 Å for Particle 2). The ORR activity was directly calculated from the local volumetric strain using the relation between the ORR activity, OH binding energy, and local volumetric strain for each facet type. To obtain the final ORR-strain relation, we need to combine the [ORR]-[OH binding energy] relation and the [OH binding energy]-[strain] relation. The relations between the ORR and OH binding energy can be found from literature[22,51,52] (Supplementary Fig. 24a). The relations between the OH binding energy and applied strains for {111} and {100} facets are also available[14,50]. For {110} facets, we performed density functional theory (DFT) calculations to obtain the relation between the OH binding energy and strain. The Vienna ab initio simulation package (VASP)[65] was used with the Perdew, Burke, and Ernzerhof (PBE)[66] exchange-correlation functional and the projector-augmented wave (PAW)[67,68] pseudopotentials. The kinetic energy cutoff of 450 eV and no spin-polarization were applied. The adsorption energies of OH were calculated using a $2 \times 2 \times 1$ supercell for {110} facet with four layers of which the two bottom ones are frozen. The $8 \times 8 \times 1$ Monkhorst-Pack $k$-grid was used for slab structure. A vacuum of 16 Å was applied to avoid the spurious interaction between the slabs along the $z$-direction and the structure is optimized until forces are below 0.01 eV/Å. The strain was applied symmetrically to the $x$- and $y$-direction within the range from −3% to +3% with 0.5% intervals. The gas-phase reference of OH is referenced to $H_2O$ and $H_2$. In the calculation, the OH was located at the top site of the surface atom. We also confirmed the OH binding energy for the fcc site of an unstrained Pt {111} facet with a $3 \times 3 \times 1$ supercell, which is a reference point for our ORR calculation. The strain-dependent OH binding energies for {110} facets were fitted with a quadratic function (Supplementary Fig. 24b), resulting in the final ORR-strain relation for the {110} facets (Supplementary Fig. 24c). The assigned facet, local lattice constant, volumetric strain, and ORR activity of each atom are illustrated in Supplementary Movies 4 and 8.

## MS simulation for core-shell structure

Molecular statics simulations were conducted via the LAMMPS package[69]. There are a few EAM potentials[53,54] for the Pd-Pt system, but they are not appropriate for binary compounds. Besides, they cannot incorporate the lattice constant inversion due to the hydrogenation-induced Pd lattice expansion. Moreover, in terms of the Zener anisotropy ratio $A = 2C_{44}/(C_{11} - C_{12})$[70] listed in Supplementary Table 1, Pt is reported to have relatively small elastic anisotropy of $A = 1.6$[71] (note that $A = 1$ means an elastically isotropic material). Since the strain distribution is affected by the elastic anisotropy, we chose Al instead of Pt when performing MS simulations. As shown in Supplementary Table 1, existing EAM potentials do not properly predict the anisotropy ratio of Pt[72], while the predicted anisotropy ratio of Al is closest to the experimentally measured anisotropy of Pt. Then, Ag is chosen to represent the core because the lattice constant ratio of Ag (4.09 Å) to Al (4.05 Å) is similar to the ratio of hydrogenated Pd (3.92 Å) to Pt (3.87 Å). We created 4 initial configurations for the simulations. The first two are the ideal fcc lattice-based core-shell models fitted to the experimental atomic structures of Particle 1 and Particle 2, respectively. In addition, we made an ideal spherical core-shell structure consisting of Ag core with a radius of 30 Å and Al shell with a thickness of 20 Å. Furthermore, to mimic the exposed core in the experimentally measured Particle 1, we partially cut the perfectly spherical core-shell at the (111) plane 30 Å away from the center of the core. We used the conjugated gradient (CG) method for energy minimization and an EAM potential (https://sites.google.com/site/eampotentials/AlAg) for Ag-Al interaction. Finally, we obtained the 3D strain maps of the Ag@Al structures using the same procedure of strain calculation as described above. The reference lattice constants for the displacement and strain

calculation were obtained to be 4.04 Å for the experimental model of Particle 1, 4.03 Å for the experimental model of Particle 2, 4.04 Å for the perfectly spherical model, and 4.04 Å for the partially cut spherical model, respectively.

## EDS experiment

An energy-dispersive X-ray spectroscopy (EDS) analysis was conducted using a field emission transmission electron microscope (FEI Talos F200X) operated at 200 kV at the KAIST Analysis Center for Research Advancement (KARA). The specimen was prepared by depositing a solution of the synthesized Pd@Pt nanoparticles dispersed in water onto a 5 nm thick silicon nitride membrane grid, followed by a vacuum annealing at 120 °C for 24 h. The elemental mapping images were acquired in ADF-STEM mode by a Super-X 4 windowless SDD EDS detector, and the size of the images was $512 \times 512$ pixels with a pixel size of 32.26 pm (Supplementary Fig. 1a–c) and 16.26 pm (Supplementary Fig. 1d–f).

## The volumetric strain calculation from XAFS experiment data

In a previous study[49], Pd@Pt core-shell nanoparticles similar to those used here were synthesized and the interatomic distances for each element were obtained by XAFS fitting. From the interatomic distances, the volumetric strain was calculated as $(a_{XAFS} - a_{ref})/a_{ref}$, where the $a_{ref}$ is the bulk lattice constant for each element[39,41] and $a_{XAFS}$ is the lattice constant calculated from the obtained interatomic distance for each element. The averaged volumetric strains for all Pd@Pt core-shell nanoparticles from the XAFS experiments are 1.6% for Pd and −0.3% for Pt.

## Experimental analysis for ORR activity

For the experimental analysis of ORR activity according to the particle size, nanoparticle samples with precisely controlled sizes were synthesized by modulating the ramping rate or the amount of polyvinylpyrrolidone. To synthesize Sample 1 which is similar in size to Particle 1, the ramping rate for heating (up to 200 °C) was adjusted to 20 °C min⁻¹. In the case of Sample 2 which is similar in size to Particle 2, the amount of polyvinylpyrrolidone was increased to 200 mg and the ramping rate was adjusted to 20 °C min⁻¹. Other experimental conditions were the same as those employed in the original synthesis. The ORR polarization curves were obtained by linear sweep voltammetry in $O_2$-saturated 0.1 M $HClO_4$ solution at a scan rate of 10 mV s⁻¹ with 1600 rpm rotation rate using CH Instruments model 760D. A glassy carbon electrode (diameter = 5 mm), Pt wire, and reversible hydrogen electrode (RHE, hydroflex reference electrode, Gaskatel) were used for the working electrode, counter electrode, and reference electrode, respectively. Ohmic iR compensation was applied to the ORR polarization curves. The following Koutecky-Levich equation was used to calculate the kinetic current density ($j_k$)[73]

$$1/j = 1/j_k + 1/j_d \qquad (2)$$

where $j_d$ is the diffusion limit current density.

## Reporting summary

Further information on research design is available in the Nature Research Reporting Summary linked to this article.

# Data availability

Source data for dot/scatter/line plots are provided with this paper. All of our experimental data, tomographic reconstructions, determined atomic structures, and simulated atomic structures will be posted on a public website (http://mdail.kaist.ac.kr/coreshell), and they can also be accessed through an open repository (https://doi.org/10.5281/zenodo.7040936) upon publication.

## Code availability

Source codes will be posted on a public website (http://mdail.kaist.ac.kr/coreshell), and they can also be accessed through an open repository (https://doi.org/10.5281/zenodo.7040936) upon publication.

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

## Acknowledgements

We thank Chang Yun Son and Jaewhan Oh for helpful discussions. This research was mainly supported by Samsung Science and Technology Foundation (SSTF-BA2201-05). Nanoparticle synthesis and ORR measurements were supported by the National Research Foundation of Korea (NRF) Grants funded by the Korean Government (MSIT) (2015R1A3A2033469, 2018R1A5A1025208). H.J. C.J. and J.L. were also partially supported by the KAIST-funded Global Singularity Research Program (M3I3) for 2019, 2020, and 2021. The EDS analysis was conducted using a field emission transmission electron microscope (FEI Talos F200X) in the KAIST Analysis Center for Research Advancement (KARA). Excellent support by Ji Eun Lim and the staff of KARA is gratefully acknowledged. The experiment utilizing the TEAM 0.5 microscope was performed at the Molecular Foundry, which is supported by the Office of Science, Office of Basic Energy Sciences of the US Department of Energy under contract no. DE-AC02-05CH11231. Y. -L. Lee would like to acknowledge the support from KISTI supercomputing center through the strategic support program for the supercomputing application research (No. KSC-2021-CRE-0144).

## Author contributions

Y.Y. conceived the idea and directed the study. D.H.W. and S.W.H. synthesized the Pd@Pt nanoparticles. H.J., H.B., A. J. P., W.T., P.E. and Y.Y. performed the electron tomography experiment. T.L. and S.R. per-formed the molecular statics simulations. Y.K. performed the oxygen reduction reaction activity measurement experiment. Y.-L.L. performed the density functional theory calculation. H.J., C.O., C.J., J.L., and Y.Y conducted the experimental data analysis of electron tomography. H.J., D.H.W., T.L., Y.K., S.R., S.W.H., and Y.Y. wrote the manuscript. All authors commented on the manuscript.

## Competing interests

The authors declare no competing interests.
