## [Peer Review File · Nature Communications]

Reviewer comments , first round review -

Reviewer #1 (Remarks to the Author):

This manuscript reported an advanced atomic electron tomography (AET) technique to exquisitely elucidate three-dimensional (3D) atomic structure and strain configuration of Pd@Pt core-shell nanoparticles with different size and shell thickness. Several conclusions were explicitly obtained via this powerful technology, including: (1) For larger nanoparticles with diameter of 8 nm or so, partial core element atoms may exist on the surface and exhibit remarkable strain effect (tensile effect for Pd atoms in this work); (2) Misfit of lattice and thus intrinsic strain can be observed both on the surface and along the interface, the relationship between which suggests certain linear tendency and can be used to mutual prediction and tuning; (3) The prominent anisotropy of strain displacement shall be attributed to the core-shell geometry and the shape of the nanoparticles, which has nothing to do with specific facets.

This tomography technique enables intuitive visualization of 3D atomic distribution and can be applied as a powerful tool for strain effect analysis towards deeper study of structure-performance relationship and rational design of nanocrystals in catalysis field.

1. Please highlight the conclusions at the end of the article, like the first paragraph in this review. Such a summary helps easier glimpse of the main content of the research and may contribute to more reading and attention from other researchers.
2. Please add more contrast samples if possible, at least a Particle 3 with similar size as Particle 1 and same shell thickness as Particle 2 so that the size effect and shell-thickness effect can be experimentally distinguished. Otherwise, at least more detailed discussion or calculation is needed to clarify this issue, which remains a little bit ambiguous for current. Moreover, series of nanoparticles with same diameter but different shell thickness are greatly expected to be studied via AET since it may bring about more inspiration and is more instructive in practical fabrication and catalysis areas.
3. Please provide simple illustration of AET mechanism and prediction or discussion about its potential application in in-situ image characterization of nanocrystal electrocatalysts, which is much more significant and useful in relevant area. If we can use AET for in-situ observation of the nucleation and growth of nanocrystals and its dynamic transformation during electrochemical process, amazing information should be got and this possibility is highly expected to be discussed or at least, mentioned in this article to gain more research attention. If there exist some factors that hinder its utilization in in-situ application, please elucidate them so that readers can learn more about this new technology.
4. Please compare the micro-area strain qualification outcome from AET to its integral counterpart from XAFS fitting. Discussion about the relationship between these two characterizations outcomes and the pros and cons of each method is welcomed to be displayed in order to provide inspiration for further study focused on strain effect.
5. Please give out the experimental data of ORR performances of the two core-shell nanoparticles and compare it with the DFT-predicted outcomes.

Reviewer #2 (Remarks to the Author):

Revealing 3-dimensional core-shell interface structures at the single-atom level by JO et. al. is an interesting study, but needs major revision before publication.

1. Strain-dependent ORR activity and histogram of surface volumetric strain, why DFT and counts are matching for 111 but not for 100 facet
2. Author must justify clearly the reason for “we classified the surface atoms into the two dominant facet families {111} and {100}”, why 110 is ignored.
3. Author must justify the following line-“We expect that this understanding will pave a new way toward the development of long-desired low-cost catalysts with high efficiency”.. Pt and Pd study and low cost are not correlated.
4. Author must highlight importance of Pt-Pd study in introduction and cite references e.g. ACS Catalysis 10 (6), 3658-3663,2020; Nanoscale 12 (22), 11830-11841,2020; Nanoscale 10 (18), 8840-8850,2018; ACS Catalysis 11 (22), 14000-14007,2021; Materials Today Energy 16, 100393,2020.
5. Author are requested to just show interface (two-three-unit cell both side) of Pt-Pd and show their relative orientation.

Reviewer #3 (Remarks to the Author):

The manuscript describes atomic resolution measurement of 3D strain in nanoparticles, compares the experimental results to molecular statics simulation and calculates oxygen reduction reaction activities of Pt surface in a similar way as in Ref. 43. The novelty of this manuscript is the application to a core-shell nanoparticles, proving Poisson effect on the atomic scale, making conclusions on the role of elastic anisotropy and proving correlation of the interfacial strain to surface strain. The work is of a high interest for materials science (especially strain engineering). The work is of overall high quality, the methods are described in a detail, the experiments were carefully conducted (e.g. checking the radiation damage) and data were carefully evaluated (e.g. missing wedge elongation corrected when atom tracing). I recommend the publication in Nature Communications. However, I suggest few improvements.

Concerning atomic electron tomography, the authors should cite not only their own papers (Ref. 29 - 31) but also another group, B. Goris et al, "Measuring Lattice Strain in Three Dimensions through Electron Microscopy", Nano Lett. 2015, 15, 10, 6996–7001, <https://pubs.acs.org/doi/10.1021/acs.nanolett.5b03008>

Sentence: "we measured tomographic tilt series for two nanoparticles: one with a diameter of 8.56 nm (Particle 1) and another with a diameter of 5.76 nm (Particle 2) (Methods)."
Both particles are odd shaped and the classification by diameter only is not appropriate.

Concerning practical application of 3D strain mapping with atomic resolution the authors should

mention the time spent on acquisition and data analysis, especially since hundreds of atoms had to be manually added or removed. ("To finalize the 3D atomic structure, a manual correction was applied to add (or remove) physically (un)reasonable atom candidates. ... Total 749 (Particle 1) and 489 (Particle 2) atoms were manually added, and 899 (Particle 1) and 771 (Particle 2) atoms were manually removed.")

The EDS of the nanoparticle and Suppl. Fig. 1 needs clarification:

"The EDS map (Supplementary Fig. 1) clearly shows the Pd signal near the surface, evidencing that the observed surface Pd atoms are real."

On the contrary to the tomography of the two particles, the EDS in Suppl. Fig. 1 is not recorded at atomic resolution. The elemental maps have low resolution and are too noisy to conclude from them there is a Pd monolayer on the Pt shell (compare to Fig. 1 d-g and Suppl Fig. 8 d-g). It is not stated if the elemental maps are raw data or background-corrected data. The authors should explicitly state that the particle in Suppl. Fig. 1 is neither Particle1 nor Particle2 but a third particle.

Comparing Particle1 and Particle2: The authors write "the Pt surface lattice is not fully relaxed due to a thinner Pt shell thickness" but Fig. 3d and Suppl. Fig 11d both show 2 nm Pt shell. The difference between the particles is in the size of the core, not in the thickness of the shell.

On MS simulations: The Fig 2de could be compared side-by-side with Fig 4bc to better see their matches and differences. Now there are 20 slices in Fig. 2 and 10 slices in Fig. 4.

The surface and interface strain in disturbed (shell not continuous) Particle1 seems to be better correlated (Fig. 3c) than in undisturbed Particle2 (Suppl. Fig. SI 11c). Authors could comment on this and also on the slope of the fitted linear regression. It is not stated if the Pd atoms on the exposed surface of the Particle1 are included.

Suppl. Fig. 4 "ORR versus strain" is referenced in main text but the significance of the histogram of the volumetric strain is not commented in the main text. Suppl Fig. 13 is not referenced nor commented in the main text. Are Pd atoms at the exposed part of the Particle1 excluded from the histogram? The authors could explain differences in histograms in Suppl. Fig.4 and Suppl. Fig. 13.

The identification of atom coordinates is described in detail in the Methods part, but identification of the chemical species of the atoms is only stated in one sentence "The atoms at finalized atomic positions were classified into Pd or Pt chemical species by using the classification method based on the k-means clustering as described in the previous works^{30,31}." The authors could briefly describe how the atom species were distinguished and add the histogram of the identified local intensity peaks like in Ext.Fig. 2a in Ref. 30. Ref. 30 compared Fe and Pt atoms where the contrast is expected to be much higher than in case of Pd versus Pt atoms in this manuscript. The histogram would show the readers how easy or difficult is to distinguish the intensities of Pd and Pt.

The authors could briefly comment if there are some essential differences in the data acquisition and data processing compared to their paper Ref 43. The Ref. 43 also measures strain and calculates ORR.

SupplVideo2 and SupplVideo6 from "Zip of files for Reviewer" have low resolution and are highly compressed. It is not possible to distinguish blue and black dots.

In SupplVideo4 and SupplVideo8 could have a legend of meaning of colors of the upper left particle. (It is described in the suppl. info, but label in the video would make it easier for reader to understand the video.)

First, we thank the referees for taking the time and effort to provide constructive comments, which have been very helpful in making improvements to the manuscript.

Please find below our response to the reviewers' comments on our manuscript "*Revealing 3-dimensional core-shell interface structures at the single-atom level*". We have carefully addressed all the referees' criticisms and suggestions and trust that with the presented changes, the manuscript is now acceptable for publication in *Nature Communications*.

Response to reviewers' comments

Reviewer #1:

This manuscript reported an advanced atomic electron tomography (AET) technique to exquisitely elucidate three-dimensional (3D) atomic structure and strain configuration of Pd@Pt core-shell nanoparticles with different size and shell thickness. Several conclusions were explicitly obtained via this powerful technology, including: (1) For larger nanoparticles with diameter of 8 nm or so, partial core element atoms may exist on the surface and exhibit remarkable strain effect (tensile effect for Pd atoms in this work); (2) Misfit of lattice and thus intrinsic strain can be observed both on the surface and along the interface, the relationship between which suggests certain linear tendency and can be used to mutual prediction and tuning; (3) The prominent anisotropy of strain displacement shall be attributed to the core-shell geometry and the shape of the nanoparticles, which has nothing to do with specific facets.

This tomography technique enables intuitive visualization of 3D atomic distribution and can be applied as a powerful tool for strain effect analysis towards deeper study of structure-performance relationship and rational design of nanocrystals in catalysis field.

1. *Please highlight the conclusions at the end of the article, like the first paragraph in this review. Such a summary helps easier glimpse of the main content of the research and may contribute to more reading and attention from other researchers..*

Response:

We thank the reviewer for providing a very detailed suggestion that can significantly improve the readability of our manuscript. Based on the suggestion, we rewrote the conclusion parts, which now read "First, partial exposure of the core part on the surface resulted in a remarkable tensile strain effect in the local region for the larger nanoparticle. Second, the misfit of lattice and thus intrinsic strain can be observed both on the surface and along the interface, and the relationship between them suggests a linear tendency. Third, the surface displacement behavior clearly demonstrated the Poisson effect globally as well as locally. Fourth, the prominent anisotropy of strain displacement shall be attributed to the core-shell geometry and the shape of the nanoparticles, rather than surface facet effects, as confirmed by MS simulations."

2. *Please add more contrast samples if possible, at least a Particle 3 with similar size as Particle 1 and same shell thickness as Particle 2 so that the size effect and shell-thickness effect can be experimentally distinguished. Otherwise, at least more detailed discussion or calculation is needed to clarify this issue, which remains a little bit ambiguous for current. Moreover, series of nanoparticles with same diameter but different shell thickness are greatly expected to be studied via AET since it may bring about more inspiration and is more instructive in practical fabrication and catalysis areas.*

Response:

We thank the reviewer for very constructive suggestions. As suggested by the reviewer, we comprehensively studied the thickness effect in more detail to clarify the difference we observe between the two particles (Particle 1 and Particle 2). Since our technique can identify the 3D locations of thousands of shell atoms individually, the lattice relaxation effect can be revealed as a function of the distance from the interface. From Revision Fig. 1, it can be seen that the in-plane lattice constant gradually decreases as the distance from the interface increases for both Particle 1 and Particle 2, showing a clear lattice relaxation behavior toward the surface. Note that Particle 2 shows a relatively larger lattice constant at the interface (due to the relatively larger lattice constant of the Pd core for Particle 2), and also exhibits a faster rate of lattice relaxation compared to Particle 1.

To compare the thicknesses of different particles, the shell thicknesses for each interface atom-surface atom pair were averaged for each particle ('Determination of the interface and surface atom pairs' part of Methods). Considering the average shell thickness of 9.75 Å for Particle 1 and 6.77 Å for Particle 2, this analysis demonstrates that the relatively larger local surface lattice constants of the Pt shell observed for Particle 2 can be (at least partially) attributed to the thickness effect; since the lattice constant decreases as a function of the distance from the interface, the surface lattice constant is expected to be smaller for a thicker shell. However, as can be seen from the differences in the interface lattice constants and lattice relaxation rates, other effects (the overall shape, size, larger core lattice constant [this might be due to the different degree of hydrogenation of the Pd], etc.) can also affect the behavior of the surface lattice constant.

We expect that a full understanding of the size and thickness effect can be achieved via controlled AET experiments by systematically adjusting the particle shape, core size, shell thickness, and the degree of hydrogenation of the Pd. As the reviewer suggested, this will bring valuable information to the field of nanocrystal fabrication and catalysis. However, it will require tremendous development and optimization not only in AET measurement and analysis, but also in nanocrystal synthesis, which is beyond the scope of this paper (here we focus on surface-interface strain effect, Poisson effect, and physical properties [e.g. ORR activity] calculations through strain analysis). We plan to cover the size-thickness effect in the follow-up studies, for which we are developing new shape-controlled synthesis methods that can prevent the hydrogenation of the Pd core.

We added this discussion in the first paragraph of 'The behavior of the Particle 2' and the conclusion part of the manuscript, which now read "From Supplementary Fig. 20, it can be seen that the in-plane lattice constant gradually decreases as the distance from the interface increases for both Particle 1 and Particle 2, showing a clear thickness-dependent lattice relaxation behavior. Considering the average shell thickness of 9.75 Å for Particle 1 and 6.77 Å for Particle 2 (Methods and Supplementary Fig. 25), this analysis demonstrates that the relatively larger local surface lattice constants of the Pt shell observed for Particle 2 can be (at least partially) attributed to the thickness effect; since the lattice constant decreases as a function of the distance from the interface, the surface lattice constant is expected to be smaller for a thicker shell. However, as can be seen from the differences in the interface lattice constants and lattice relaxation rates, other effects (the overall shape, size, larger core lattice constant [this might be due to the different degree of hydrogenation of the Pd], etc.) can also affect the behavior of the surface lattice constant.", and "Looking forward, controlled AET experiments via fine-tuning of the particle shape, core size, shell thickness, and the degree of hydrogenation of the Pd can shed light on the size and thickness effect, which can bring valuable information in the field of nanocrystal fabrication and catalysis.", respectively, and included the interface distance dependent lattice relaxation figure as Supplementary Figure 20.

Revision Figure 1 | Radial profile of the in-plane lattice constant. a-b, The in-plane lattice constant plotted as a function of the distance from the interface for Particle 1 (a), and Particle 2 (b). Note that Particle 2 shows a relatively larger lattice constant at the interface (due to the relatively larger lattice constant of the Pd core for Particle 2), and also exhibits a faster rate of lattice relaxation compared to Particle 1. The in-plane lattice constants were estimated from the averaged in-plane nearest neighbor distance. The in-plane nearest neighbors for a given atom are defined as the nearest neighbor atoms for which the line connecting the given atom and the nearest neighbor atom makes an angle below 22.5° to the plane perpendicular to the position vector of the given atom. Each data point and error bar represent an average and standard deviation of the lattice constants within the bins of 3.5 \AA interval, respectively.

3. Please provide simple illustration of AET mechanism and prediction or discussion about its potential application in in-situ image characterization of nanocrystal electrocatalysts, which is much more significant and useful in relevant area. If we can use AET for in-situ observation of the nucleation and growth of nanocrystals and its dynamic transformation during electrochemical process, amazing information should be got and this possibility is highly expected to be discussed or at least, mentioned in this article to gain more research attention. If there exist some factors that hinder its utilization in in-situ application, please elucidate them so that readers can learn more about this new technology.

Response:

We thank the reviewer for very helpful and interesting comments. We fully agree on the importance of *in-situ* image characterization in the field of nanocrystal electrocatalysts. As materials research advances, it is becoming increasingly important to observe the dynamics of nanomaterials in real conditions. There are some obstacles that should be addressed to properly implement the *in-situ* capability into AET (such as sample durability, time resolution, and functional limitations of the *in-situ* tomography holder), but we expect that we will be able to find solutions based on some recent advancements in the field¹⁻⁴.

We added this discussion to the conclusion part of the manuscript, which now read “Furthermore, if *in-situ* capability can be properly implemented in the AET by overcoming some technical obstacles (e.g., sample durability, time resolution, and TEM holder design), it will allow us to unveil the 3D atomic dynamics of nanomaterials during dynamical processes such as segregation/diffusion, sintering, and device operation.”, and we included an illustration of the AET mechanism (Revision Fig. 2) as Supplementary Figure 21, as the reviewer suggested.

Revision Figure 2 | A schematic layout of atomic electron tomography based on ADF-STEM. a, An electron beam is focused on a small spot by electromagnetic lenses, and scanned over a specimen while the integrated signal at each scanning position is recorded by an annular detector to form a 2D projection image. **b,** A series of 2D projections are measured at different sample tilt angles. **c,** After image post-processing, the tilt series becomes converted to Fourier slices and a 3D reconstruction can be computed by using an iterative algorithm (for example, GENFIRE⁵).

4. *Please compare the micro-area strain qualification outcome from AET to its integral counterpart from XAFS fitting. Discussion about the relationship between these two characterizations outcomes and the pros and cons of each method is welcomed to be displayed in order to provide inspiration for further study focused on strain effect.*

Response:

We thank the reviewer for raising an interesting point. To compare with XAFS results which only provide an averaged volumetric strain information, we averaged the volumetric strains of each atom obtained from the AET results and compared them with the previously reported strain results obtained by XAFS study⁶ of a similar Pd@Pt system. In the study⁶, Pd@Pt core-shell nanoparticles similar to those used here were synthesized and the interatomic distances for each element were obtained by XAFS fitting. From the interatomic distances, the volumetric strain was calculated as $(a_{XAFS} - a_{ref})/a_{ref}$, where the a_{ref} is the bulk lattice constant for each element^{7,8} and a_{XAFS} is the lattice constant calculated from the obtained interatomic distance for each element. The averaged volumetric strains for all Pd@Pt core-shell nanoparticles from the XAFS experiments are 1.6% for Pd and -0.3% for Pt. Averaging the volumetric strain obtained from AET for every atom in the two nanoparticles yields volumetric strains of 2.1 % for Pd and -0.4 % for Pt. Both of the results from XAFS and AET consistently show relatively large tensile strains at the core and mild compressive strains in the shell. XAFS experiments and analyses are relatively well-established and sample a wide range of nanoparticle sizes to obtain the average strain information of the target system. However, it can only provide averaged information and is blind to any local atomic-scale details which depend on core size, shell thickness, shape, composition, and/or facet distribution. AET is still under rapid development and currently can only be applied to a few nanoparticles rather than providing global information of nanoparticle ensembles, but it can detect individual atom level details regarding the core-shell system and provide detailed local information including local strain

distribution, full 3D strain profile for the entire core-shell structure, distribution of surface volumetric strain and related local ORR activity, and the mixed effects of thickness/shape/facet at the single-atom scale throughout each individual nanoparticles.

We have included this discussion in the fifth paragraph of ‘Experimental identification of core-shell 3D atomic structure’ part of our revised manuscript and ‘The volumetric strain calculation from XAFS experiment data’ part in the method section, which now read “The averaged volumetric strains can be calculated from the local lattice constants, which can be compared with those from a previous X-ray absorption fine structure (XAFS) study of a similar Pd@Pt system. The averaged volumetric strains of Pd@Pt core-shell nanoparticles from the XAFS experiments are 1.6% for Pd and -0.3% for Pt. Averaging the volumetric strain obtained from AET for every atom in the two nanoparticles yields volumetric strains of 2.1 % for Pd and -0.4 % for Pt. Both of the results from XAFS and AET consistently show relatively large tensile strains at the core part and mild compressive strains in the shell. XAFS experiments sample a wide range of nanoparticle sizes to obtain the average strain information of the target system to provide averaged information and are blind to any local atomic-scale details. AET currently can only be applied to a few nanoparticles rather than providing global information of nanoparticle ensembles, but it can detect individual atom level details regarding the core-shell system and provide detailed local information at the single-atom scale throughout each individual nanoparticle.” and “In a previous study, Pd@Pt core-shell nanoparticles similar to those used here were synthesized and the interatomic distances for each element were obtained by XAFS fitting. From the interatomic distances, the volumetric strain was calculated as $(a_{XAFS} - a_{ref})/a_{ref}$, where the a_{ref} is the bulk lattice constant for each element and a_{XAFS} is the lattice constant calculated from the obtained interatomic distance for each element. The averaged volumetric strains for all Pd@Pt core-shell nanoparticles from the XAFS experiments are 1.6% for Pd and -0.3% for Pt.”, respectively.

5. *Please give out the experimental data of ORR performances of the two core-shell nanoparticles and compare it with the DFT-predicted outcomes.*

Response:

We thank the reviewer for a valuable suggestion. Unfortunately, it is almost impossible to measure ORR performance for a single particle due to experimental limitations. In this regard, we prepared two different nanoparticle samples, Samples 1 and 2, which have average sizes similar to the sizes of Particles 1 and 2, respectively, and investigated their ORR activity. To synthesize the nanoparticle sample sets of narrower size distributions, we modified the synthesis method by adjusting the ramping rate and the amount of polyvinylpyrrolidone (‘Experimental analysis for ORR activity’ part in the Methods). The ADF-STEM images of the prepared Samples 1 and 2 are shown in Revision Fig. 3a,b. The average sizes of Samples 1 and 2 were 8.51 and 5.69 nm, respectively, which are very similar to the average diameters of Particle 1 (8.56 nm) and Particle 2 (5.76 nm). According to EDS mapping images (Revision Fig. 3c,d), the Pd core-Pt shell structure could be confirmed. The synthesized samples were deposited on a carbon black support for ORR measurements (Revision Fig. 3e,f).

The ORR polarization curves were obtained in O₂-saturated 0.1 M HClO₄ at a scan rate of 10 mV s⁻¹ (Revision Fig. 4). The half-wave potential of Sample 1, 0.848 V vs. RHE, is higher than that of Sample 2, 0.824 V vs. RHE, indicating that the ORR activity of Sample 1 is higher than that of Sample 2. For more precise comparison, specific activities were calculated by normalizing the kinetic current values, which were obtained from the Koutecky-Levich equation, with respect to the electrochemically active surface area at 0.9 V vs. RHE; the specific activities of Samples 1 and 2 were 0.24 and 0.063 mA cm⁻², respectively. Based on these experimental results, we can confirm

that the ORR activity of Sample 1 is superior to that of Sample 2, which is in good accordance with the conclusion from the DFT-predicted outcomes.

We added this method to the ‘Experimental analysis for ORR activity’ part in the method section. Also, we have included this discussion in the first paragraph of ‘The behavior of the Particle 2’ part of the manuscript which now reads “Experimental ORR activity measurements were also performed on two groups of nanoparticles with similar sizes to Particle 1 and Particle 2 (Methods and Supplementary Figs. 22 and 23). From Supplementary Fig. 23, it can be seen that the group of particles having a size similar to Particle 1 shows a better ORR activity than the group with the size similar to Particle 2.”, and included the Revision Figure 3 and 4 as Supplementary Figure 22 and 23, respectively.

Revision Figure 3 | Structural and elemental analysis of newly synthesized samples for ORR measurements. a-b, ADF-STEM images of the two groups of synthesized Pd@Pt nanoparticles, for the group with the average diameter 8.56 nm [Sample 1] (a) and that with the average diameter 5.76 nm [Sample 2] (b). **c-d**, EDS elemental mapping images correspond to (a) and (b), respectively. **e-f**, Bright-field TEM images of the samples supported on a carbon black, for Sample 1 (e) and Sample 2 (f).

Revision Figure 4 | ORR polarization curves of samples in O₂-saturated 0.1 M HClO₄ at a scan rate of 10 mV s⁻¹.

Reviewer #2:

Revealing 3-dimensional core-shell interface structures at the single-atom level by JO et. al. is an interesting study, but needs major revision before publication.

1. *Strain-dependent ORR activity and histogram of surface volumetric strain, why DFT and counts are matching for 111 but not for 100 facet.*

Response:

We thank the reviewer for raising an interesting point. In Supplementary Fig. 4, the histogram represents the distribution of experimentally determined surface volumetric strain for each facet, and the DFT result (orange line) merely relates the expected ORR and local strain. By coincidence, the most probable surface strain (obtained from the experiment) and the strain which gives the maximum ORR (the peak of the DFT-calculated volcano curve) match for {111} facets, but there is no reason for the experimental strain distribution to be centered at the maximum ORR strain point. For another particle (Particle 2) in Supplementary Fig. 13, it can be seen that the experimental strain distribution is not exactly centered around the maximum ORR strain points both for {100} and {111} facets.

To make this point clear, additional comments have been added to the captions of Supplementary Figs. 4 and 13, which now read “The DFT result (orange lines) is provided to show the calculated relation between the ORR and local strain.”

2. *Author must justify clearly the reason for “we classified the surface atoms into the two dominant facet families {111} and {100}”, why 110 is ignored.*

Response:

We thank the reviewer for raising an important point. To obtain the final ORR-strain relation, we need to combine the “[ORR]-[OH binding energy relation]” and the “[OH binding energy]-[strain relation]”. The reason that only the {100} and {111} facets were initially considered is that the relation between the OH binding energy and strain for {100} and {111} facets was provided in the previous study⁹. However, to address the reviewer’s important concern, we additionally performed density functional theory (DFT) calculations to obtain the relation between the OH binding energy and strain for the Pt {110} facet. Since the relation between the ORR activity and the OH binding energy for the Pt {110} facet is already known¹⁰ (Revision Fig. 5a), combined with our DFT calculations for “[OH binding energy]-[strain] relation” (Revision Fig. 5b), we were able to obtain the ORR-strain relation for the {110} facet (Revision Fig. 5c). From the plots, we found that OH binding energies for Pt {110} facet are less sensitive to the applied strain than {100} and {111} surfaces.

Next, we classified the surface atoms into the three dominant facet families of {111}, {100} and {110} this time (‘ORR activity calculation at the surface’ part in the Methods). The ORR activity was directly calculated from the local volumetric strain of surface atoms using the obtained ORR-strain relation for each facet type. Revision Figure 6 represents the revised pole figures of the surface ORR activity for the upper and lower hemispheres for Particle 1 (Revision Fig. 6a) and Particle 2 (Revision Fig. 6b).

We revised Fig. 3e,f, Supplementary Figs. 4,11,13, and Supplementary Videos 4,8 of the manuscript accordingly, and the Revision Fig. 5 is now the Supplementary Figure 24 in our revised manuscript. We also included detailed information about the ORR activity calculation for {110}

facets in the ‘ORR activity calculation at the surface’ part in the method section of our revised manuscript.

Revision Figure 5 | The relation between ORR activity, OH binding energy, and strain. **a**, Volcano-shaped relations between the ORR activity and OH binding energy for Pt {111}, {100} and {110} facets. The OH binding energy in the x-axis represents the relative change of the OH binding energy (E_b) with respect to the unstrained surfaces (E_b^0). **b**, Strain-dependent relative change of the OH binding energy for {111} and {100} facets (blue and red solid lines, respectively, obtained from a previous study¹¹) and that for {110} facet (green circles; obtained from our own density functional theory calculation). The green solid line represents the fitted quadratic function for the green data points. **c**, The final relations between the ORR activity and strain for the three facet types obtained by combining the relations in (a) and (b). The ORR activity is represented as $\ln(j/j_{Pt(111)})$ where j is the current density.

Revision Figure 6 | Pole figures of the surface ORR activity. a-b, Pole figures obtained from stereographic projections of the surface ORR activity for the upper and lower hemispheres, for Particle 1 (**a**) and Particle 2 (**b**). The ORR activity is represented as $\ln(j/j_{Pt(111)})$ where j is the current density. The Miller indices of low-index facets are marked at the average position of the surface atoms assigned to each facet (Methods).

Revision Figure 7 | Strain-dependent ORR activity and histogram of surface Pt volumetric strain (Particle 1). a-c, The surface ORR activity as a function of the surface strain, obtained from DFT calculation (orange lines), and histogram of the surface Pt volumetric strain, for {100} facets (a), {111} facets (b), and {110} facets (c), respectively. The ORR activity is represented as $\ln(j/j_{Pt,(111)})$ where j is the current density. Note that the Pd atoms on the exposed surface are not included in this analysis. The DFT result (orange lines) is provided to show the calculated relation between the ORR and local strain.

Revision Figure 8 | Strain-dependent ORR activity and histogram of surface Pt volumetric strain (Particle 2). a-c, The surface ORR activity as a function of the surface strain, obtained from DFT calculation (orange lines), and histogram of the surface Pt volumetric strain, for {100} facets (a), {111} facets (b), and {110} facets (c), respectively. The ORR activity is represented as $\ln(j/j_{Pt,(111)})$ where j is the current density. The DFT result (orange lines) is provided to show the calculated relation between the ORR and local strain.

3. *Author must justify the following line-“We expect that this understanding will pave a new way toward the development of long-desired low-cost catalysts with high efficiency”.. Pt and Pd study and low cost are not correlated.*

Response:

We thank the reviewer for pointing this out. We agree with the reviewer, and therefore removed the ‘low cost’ part from our conclusion.

4. *Author must highlight importance of Pt-Pd study in introduction and cite references e.g. ACS Catalysis 10 (6), 3658-3663,2020; Nanoscale 12 (22), 11830-11841,2020; Nanoscale 10 (18), 8840-8850,2018; ACS Catalysis 11 (22), 14000-14007,2021; Materials Today Energy 16, 100393,2020.*

Response:

We thank the reviewer for your detailed feedback about the importance of the Pt-Pd study. As suggested by the reviewer, we highlighted the importance of Pd@Pt nanoparticles in the third paragraph of the introduction part of our revised manuscript, which now read “Here, we reveal the largely unexplored nature of a 3D core-shell interface via atomic electron tomography (AET) by using cuboctahedron-like core-shell nanoparticles with a Pd core and a Pt shell (Pd@Pt) as a model system, which has notable application potential especially in the field of catalysts.”, and cited the “*ACS Catalysis 10 (6), 3658-3663,2020; Nanoscale 12 (22), 11830-11841,2020; Nanoscale 10 (18), 8840-8850,2018; ACS Catalysis 11 (22), 14000-14007,2021; Materials Today Energy 16, 100393,2020*” papers as reference 33-37.

5. *Author are requested to just show interface (two-three-unit cell both side) of Pt-Pd and show their relative orientation.*

Response:

We thank the reviewer for the good suggestion. We agree that showing the interfacial structure would be informative for readers. As suggested by the reviewer, we added the interfacial structure (2-3 unit cells on both sides) as an inset in Fig. 1d of our revised manuscript (see Revision Fig. 9). In Revision Fig. 9, it can be clearly seen that the crystallographic directions of the Pt and Pd regions are the same; the whole nanoparticle (core + shell) structurally forms an fcc single-crystal (clearly visible from Supplementary Video 2, Fig. 1b and inset of Fig. 1d). We also briefly discussed about the fact that the particle forms an fcc single-crystal in the fourth paragraph of ‘Experimental identification of core-shell 3D atomic structure’ part of our revised manuscript.

Revision Figure 9 | Experimentally determined 3D atomic structure of a Pd@Pt core-shell nanoparticle (Particle 1). **a**, Atomic layer slices along the [001] crystallographic direction, to show the overall core-shell structure. **b**, Overall 3D view of the nanoparticle with one octant of the sphere separated to visualize the core-shell architecture. **c**, Overall 3D structure with separated core and shell. The shell is cut in half showing the core and shell interface structure. **d-g**, 1.06 Å thick slices of the 3D tomogram, showing four consecutive atomic layers along the [001] direction, respectively. The grayscale background represents the intensity of the tomogram, and the red and blue dots represent the traced atomic coordinates of Pd and Pt, respectively. The Pd and Pt atoms can be well distinguished from the intensity contrast. The inset in **(d)** shows a magnified view of the area near the interface (indicated by a yellow box). Scale bar, 1 nm.

Reviewer #3:

The manuscript describes atomic resolution measurement of 3D strain in nanoparticles, compares the experimental results to molecular statics simulation and calculates oxygen reduction reaction activities of Pt surface in a similar way as in Ref. 43. The novelty of this manuscript is the application to a core-shell nanoparticles, proving Poisson effect on the atomic scale, making conclusions on the role of elastic anisotropy and proving correlation of the interfacial strain to surface strain. The work is of a high interest for materials science (especially strain engineering). The work is of overall high quality, the methods are described in a detail, the experiments were carefully conducted (e.g. checking the radiation damage) and data were carefully evaluated (e.g. missing wedge elongation corrected when atom tracing). I recommend the publication in Nature Communications. However, I suggest few improvements.

1. *Concerning atomic electron tomography, the authors should cite not only their own papers (Ref. 29 - 31) but also another group, B. Goris et al, "Measuring Lattice Strain in Three Dimensions through Electron Microscopy", Nano Lett. 2015, 15, 10, 6996–7001, <https://pubs.acs.org/doi/10.1021/acs.nanolett.5b03008>.*

Response:

We thank the reviewer for the feedback regarding an important work to the field of atomic electron tomography. As suggested by the reviewer, we additionally cited the “B. Goris et al, *Measuring Lattice Strain in Three Dimensions through Electron Microscopy*, *Nano Lett.* 2015, 15, 10, 6996–7001,” papers as reference 32 in the third paragraph of the introduction part of our revised manuscript.

2. *Sentence: "we measured tomographic tilt series for two nanoparticles: one with a diameter of 8.56 nm (Particle 1) and another with a diameter of 5.76 nm (Particle 2) (Methods)." Both particles are odd shaped and the classification by diameter only is not appropriate.*

Response:

We thank the reviewer for raising an important point. To provide more accurate 3D shape and size information for each nanoparticle, we included the Supplementary Video 3 and 7 with proper lengthscale, and revised the first paragraph of 'Experimental identification of core-shell 3D atomic structure' part of our manuscript, which now reads “we measured tomographic tilt series for two nanoparticles: one with an average diameter of 8.56 nm (Particle 1) and another with an average diameter of 5.76 nm (Particle 2) (Methods). These nanoparticles are not fully spherical, and actual 3D shape and size information is provided in Supplementary Videos 3 and 7.”.

3. *Concerning practical application of 3D strain mapping with atomic resolution the authors should mention the time spent on acquisition and data analysis, especially since hundreds of atoms had to be manually added or removed. ("To finalize the 3D atomic structure, a manual correction was applied to add (or remove) physically (un)reasonable atom candidates. ... Total 749 (Particle 1) and 489 (Particle 2) atoms were manually added, and 899 (Particle 1) and 771 (Particle 2) atoms were manually removed.")*

Response:

We thank the reviewer for a detailed suggestion. For each particle, the tilt series measurements took about 4 hours. Image post-processing, reconstruction, tomogram post-processing, and automatic identification of atomic coordinates took about 3-4 days, and manual correction of the obtained

atomic coordinates took about 1-2 days. We added this information in the method section of the revised manuscript.

4. *The EDS of the nanoparticle and Suppl. Fig. 1 needs clarification:*

"The EDS map (Supplementary Fig. 1) clearly shows the Pd signal near the surface, evidencing that the observed surface Pd atoms are real."

On the contrary to the tomography of the two particles, the EDS in Suppl. Fig. 1 is not recorded at atomic resolution. The elemental maps have low resolution and are too noisy to conclude from them there is a Pd monolayer on the Pt shell (compare to Fig. 1 d-g and Suppl Fig. 8 d-g). It is not stated if the elemental maps are raw data or background-corrected data. The authors should explicitly state that the particle in Suppl. Fig. 1 is neither Particle1 nor Particle2 but a third particle.

Response:

We thank the reviewer for pointing out an important issue. Unlike HR-HAADF-STEM images which can resolve individual atoms on the surface, STEM-EDS analysis suffers from a poor signal-to-noise ratio and is not capable of distinguishing a single atom on a nanoparticle surface. Atomic-resolution EDS has only been demonstrated for distinguishing atomic columns which consist of multiple atoms along the electron beam direction, rather than a "single atom". Many recent reports¹²⁻²⁰ provide atomic resolution HAADF-STEM images together with poor-resolution STEM-EDS images, showing that the STEM-EDS image we provide is almost at the limit of the achievable resolution for our system. Also, please note that our EDS analysis was performed only to support the existence of the surface Pd atoms, not to resolve atomic-scale features of them. As can be seen in the manuscript Figure 1, the surface Pd atoms can form not only a monolayer on the Pt surface, but can also form multiple layers for certain facets, and we are not claiming that the Pd atoms always form a monolayer.

To further support our claim of the existence of Pd atoms at the surface, we conducted a further analysis based on our EDS data as follows. As shown in Revision Fig. 10a, we created a narrow mask (magenta area in Revision Fig. 10a) which reflects the surface shape of the particle and scanned the mask with 1.5 Å intervals along the horizontal direction (covering the yellow area marked in Revision Fig. 10a) while integrating the EDS signals within the mask. The horizontal width of the mask is 2 Å, which is similar to the (111) interplanar spacing of the measured particles. Revision Figure 10b shows the EDS signal profile for each element averaged over the mask area and the ratio (averaged Pd signal / averaged Pt signal) profile for each scan point. It can be seen that the Pd and Pt signals both increase as the mask moves towards the particle. However, for the Pd/Pt signal ratio, a clear peak can be identified at the shell surface, evidencing the presence of the surface Pd within a few angstrom ranges.

We replaced Supplementary Fig. 1 by adding the additional information of Revision Fig. 10, and we also clarified in the figure caption that the EDS measurements were conducted on a different Pd@Pt nanoparticle (not Particle 1 or 2) and that the data was smoothed (smoothing parameter 3 of ESPRIT Software from Bruker) but not background-corrected.

Revision Figure 10 | An elemental mapping image and line profiles from EDS experiments. a, An EDS elemental mapping image of a Pd@Pt nanoparticle. Scale bar, 1 nm. **b,** The averaged intensity line scan profile within the region marked with a yellow line of (a). The magenta area represents the scanned mask within which the EDS signal was averaged to draw the line profile. The triangle and square marks represent the average value of the EDS signals of the Pt and Pd atoms within the mask, respectively, and the circle marks represent the ratio of the average values from the two atomic species. A peak can be identified for the Pd/Pt signal ratio at the shell surface, evidencing the presence of the surface Pd within a few angstrom ranges. Note that the particle for this EDS analysis is a third particle, not Particle 1 or Particle 2 in the manuscript and that the data was smoothed (smoothing parameter 3 of ESPRIT Software from Bruker) without any background correction.

5. *Comparing Particle1 and Particle2: The authors write "the Pt surface lattice is not fully relaxed due to a thinner Pt shell thickness" but Fig. 3d and Suppl. Fig 11d both show 2 nm Pt shell. The difference between the particles is in the size of the core, not in the thickness of the shell.*

Response:

We thank the reviewer for raising a very good point. The average shell thicknesses of the two nanoparticles are different and the shell of Particle 2 is indeed thinner than that of Particle 1. However, due to the anisotropic shape of the nanoparticles, the maximum shell thickness of Particle 1 and Particle 2 can look similar.

In Revision Fig. 11, the histograms of shell thickness are given (see ‘Determination of the interface and surface atom pairs’ part of Methods). Although the thickness range is similar (maximum around 20 Å), the histogram of Particle 2 shows a much smaller number of atoms in the 10 Å - 20 Å range. The average shell thickness values for each nanoparticle are 9.75 Å for Particle 1 and 6.77 Å for Particle 2, respectively, indicating that the shell of Particle 2 is indeed thinner than that of Particle 1. To make this point clearer, we included the average shell thickness information in the first paragraph of ‘The behavior of the Particle 2’ part of the manuscript, and included the histogram of shell thickness as Supplementary Figure 25, and referenced the figure in the first paragraph of ‘The behavior of the Particle 2’ part of our manuscript.

Revision Figure 11 | The histograms of the shell thickness determined from the interface-surface atom pair distances. a-b, Histogram of the shell thickness for Particle 1 (**a**), and Particle 2 (**b**). The average shell thickness values are 9.75 Å for Particle 1 and 6.77 Å for Particle 2.

6. *On MS simulations: The Fig 2de could be compared side-by-side with Fig 4bc to better see their matches and differences. Now there are 20 slices in Fig. 2 and 10 slices in Fig. 4.*

Response:

We thank the reviewer for the detailed feedback. As suggested by the reviewer, we revised Fig. 4 of the manuscript and Supplementary Fig. 14 to make them consistent with Fig. 2 and Supplementary Fig. 9 (as can be seen in Revision Fig. 12 and 13).

Revision Figure 12 | 3D strain maps and pole figures of the core-shell structure obtained by an MS simulation (Particle 1). **a**, The core-shell atomic structure resulting from the MS simulation (Methods), which is divided into atomic layers along the [001] direction. Note that only one layer per every two atomic layers is plotted. The red and blue dots represent the positions of Pd and Pt atoms assigned to the fcc lattice, respectively. **b-c**, The calculated strain maps from the MS simulation result for the radial (ϵ_{rr}) (**b**) and azimuthal ($\epsilon_{\phi\phi}$) (**c**) strains in the spherical coordinate system, respectively. The strain maps were calculated from the corresponding layers presented in (**a**). **d-e**, Pole figures showing the interface and surface strains of the MS simulation result. The pole figures were obtained from stereographic projections of the radial strain at the interface and surface for the upper (**d**), and lower (**e**) hemispheres, respectively. The Miller indices of low-index facets are marked at the average position of the surface atoms assigned to each facet (Methods).

Revision Figure 13 | 3D strain maps and pole figures of the core-shell structure obtained by an MS simulation (Particle 2). **a**, The core-shell atomic structure resulting from the MS simulation (Methods), which is divided into atomic layers along the [001] direction. Note that every atomic layer is plotted. The red and blue dots represent the positions of Pd and Pt atoms assigned to the fcc lattice, respectively. **b-c**, The calculated strain maps from the MS simulation result for the radial (ϵ_{rr}) (**b**) and azimuthal ($\epsilon_{\phi\phi}$) (**c**) strains in the spherical coordinate system, respectively. The strain maps were calculated from the corresponding layers presented in (**a**). **d-e**, Pole figures showing the interface and surface strains of the MS simulation result. The pole figures were obtained from stereographic projections of the radial strain at the interface and surface for the upper (**d**), and lower (**e**) hemispheres, respectively. The Miller indices of low-index facets are marked at the average position of the surface atoms assigned to each facet (Methods).

7. *The surface and interface strain in disturbed (shell not continuous) Particle1 seems to be better correlated (Fig. 3c) than in undisturbed Particle2 (Suppl. Fig. SI 11c). Authors could comment on this and also on the slope of the fitted linear regression. It is not stated if the Pd atoms on the exposed surface of the Particle1 are included.*

Response:

We thank the reviewer for raising an important point. Relatively higher surface-interface strain correlation of Particle 1 compared to Particle 2 can be explained by observing the overall curvature of the particle surfaces. The surface of Particle 2 exhibits a larger curvature than that of Particle 1 (because it has a smaller radius). It results in larger locational deviations of the surface atoms from their lattice positions, making the correlation weaker. It can be checked by calculating the root-mean-square deviation (RMSD) between the surface atoms and their lattice positions. The RMSD values are 68.5 pm and 71.2 pm for Particle 1 and Particle 2, respectively, showing larger RMSD for the Particle 2 (which shows a larger curvature).

The slopes of the fitted linear regression for each particle are 0.87 for Particle 1 and 0.94 for Particle 2, showing similar values very close to unity for both particles. Please note that the Pd atoms on the exposed surface of the Particle 1 were not included in the interface-surface correlation analysis. We have included this discussion in the first paragraph of ‘The behavior of the Particle 2’ part of our revised manuscript and ‘Determination of the interface and surface atom pairs’ part in the method section, which now reads “Second, the surface-interface strain correlation is weaker for

Particle 2 compared to that of Particle 1. It can be attributed to the larger surface atomic deviation from the lattice positions for Particle 2, which can result from the relatively larger surface curvature (*i.e.*, smaller radius). Particle 2 shows a larger root-mean-square deviation (RMSD) between the surface atoms and their lattice positions compared to that of Particle 1 (68.5 pm and 71.2 pm RMSDs for Particle 1 and Particle 2, respectively), supporting our assertion.", and "Some atoms were categorized as both surface and interface atoms (the Pd atoms on the exposed surface of Particle 1), which were excluded from the atom pairing analysis.", respectively, and have included the information of slopes from the linear regressions in the first paragraph of 'Observation of interface-surface correlation and full strain profile' part of our revised manuscript and caption of Fig. 3 and Supplementary Fig. 11.

8. *Suppl. Fig. 4 "ORR versus strain" is referenced in main text but the significance of the histogram of the volumetric strain is not commented in the main text. Suppl Fig. 13 is not referenced nor commented in the main text. Are Pd atoms at the exposed part of the Particle1 excluded from the histogram? The authors could explain differences in histograms in Suppl. Fig.4 and Suppl. Fig. 13.*

Response:

We thank the reviewer for pointing out an important issue. Supplementary Figs. 4 and 13 are included to directly show the distribution of surface volumetric strain and related ORR activity, in addition to the graphical representation in the ORR pole figures (Fig. 3f and Supplementary Fig. 11f). As can be seen in the Revision Figs. 16 and 17, the peak of the surface strain distribution shows slightly compressive behavior for Particle 1, while that of Particle 2 is at a slightly tensile region, resulting in a higher ORR for Particle 1 in overall, as expected from the pole figures. This behavior is consistent with the observation of Particle 2 having a larger lattice constant compared to that of Particle 1. Please note that the previous version of the histograms (Revision Fig. 14 and 15) included the Pd atoms on the exposed surface, but we decided not to include those for the revised figures (Revision Figs. 16 and 17, revised Supplementary Figs. 4 and 13), since the ORR activity is related only to the Pt atoms, not Pd atoms.

We have included this discussion in the first paragraph of 'The behavior of the Particle 2' part of our revised manuscript, which now reads "Third, the enhancement of ORR activities at the {111} facets is less pronounced (Fig. 3f and Supplementary Fig. 11f: ORR pole figures). This is related to the distributions of the surface volumetric strains (Supplementary Figs. 11e-f, 13 and Supplementary Video 8); the peak of the surface strain distribution for Particle 2 shows slightly more tensile behavior compared to that of Particle 1 (consistent with the larger local lattice constant of the Pt shell of Particle 2), resulting in relatively poorer ORR activity.", which includes the comments regarding the significance of the histogram, referencing the histogram figures. Also, we replaced Supplementary Figs. 4 and 13 as shown in Revision Figs. 16 and 17, accordingly. Lastly, we clearly stated in the caption of Supplementary Fig. 4 that the Pd atoms on the exposed surface are not included in this analysis.

Revision Figure 14 | Strain-dependent ORR activity and histogram of surface Pt volumetric strain for Particle 1 (previous version). a-b, The surface ORR activity as a function of the surface strain, obtained from DFT calculation (red lines), and histogram of the surface volumetric strain, for {100} facets (a), and {111} facets (b), respectively. The ORR activity is represented as $\ln(j/j_{Pt,(111)})$ where j is the current density. Note that the Pd atoms on the exposed surface are included in this analysis.

Revision Figure 15 | Strain-dependent ORR activity and histogram of surface Pt volumetric strain for Particle 2 (previous version). a-b, The surface ORR activity as a function of the surface strain, obtained from DFT calculation (red lines), and histogram of the surface volumetric strain, for {100} facets (a), and {111} facets (b), respectively. The ORR activity is represented as $\ln(j/j_{Pt,(111)})$ where j is the current density.

Revision Figure 16 | Strain-dependent ORR activity and histogram of surface Pt volumetric strain (Particle 1). a-c, The surface ORR activity as a function of the surface strain, obtained from DFT calculation (orange lines), and histogram of the surface Pt volumetric strain, for {100} facets (a), {111} facets (b), and {110} facets (c), respectively. The ORR activity is represented as $\ln(j/j_{Pt,(111)})$ where j is the current density. Note that the Pd atoms on the exposed surface are not included in this analysis. The DFT result (orange lines) is provided to show the calculated relation between the ORR and local strain.

Revision Figure 17 | Strain-dependent ORR activity and histogram of surface Pt volumetric strain (Particle 2). a-c, The surface ORR activity as a function of the surface strain, obtained from DFT calculation (orange lines), and histogram of the surface Pt volumetric strain, for {100} facets (a), {111} facets (b), and {110} facets (c), respectively. The ORR activity is represented as $\ln(j/j_{Pt,(111)})$ where j is the current density. The DFT result (orange lines) is provided to show the calculated relation between the ORR and local strain.

9. *The identification of atom coordinates is described in detail in the Methods part, but identification of the chemical species of the atoms is only stated in one sentence "The atoms at finalized atomic*

positions were classified into Pd or Pt chemical species by using the classification method based on the k-means clustering as described in the previous works^{30,31}." The authors could briefly describe how the atom species were distinguished and add the histogram of the identified local intensity peaks like in Ext.Fig. 2a in Ref. 30. Ref. 30 compared Fe and Pt atoms where the contrast is expected to be much higher than in case of Pd versus Pt atoms in this manuscript. The histogram would show the readers how easy or difficult is to distinguish the intensities of Pd and Pt.

Response:

We thank the reviewer for the constructive comments. Our chemical species identification process is as follows.

After the manual correction of atomic positions, their chemical species (either Pt or Pd) were determined. To classify them, we used an unbiased atom classification method based on the k-means clustering algorithm²¹ as follows. (a) We calculated the integrated intensity for every atom by summing all values within a box of $3 \times 3 \times 3$ voxels centered on each rounded atom position from the 3D tomogram. Histograms of the integrated intensities for all atoms are shown in Revision Fig. 18a for Particle 1 and Revision Fig. 18d for Particle 2. Next, we set the initial threshold as the average value of the integrated intensities to initially separate the Pt (the atoms with the integrated intensity higher than the threshold) and Pd (the atoms with intensities lower than the threshold). From the initial Pt and Pd atoms, we defined the averaged intensity box of $5 \times 5 \times 5$ voxels for each species by averaging over the $5 \times 5 \times 5$ voxels centered on all atoms of the same chemical species from the 3D tomogram. (b) For each identified atom, two error functions were calculated,

$$E_{Pt} = \sum_i |P_i - A_i^{Pt}|, \quad E_{Pd} = \sum_i |P_i - A_i^{Pd}|$$

where P_i is the i -th voxel intensity, and A_i^{Pt} and A_i^{Pd} are the i -th voxel intensity of the averaged intensity box for Pt and Pd, respectively. Using the error functions, all the atoms were re-classified into Pt or Pd based on the minimal error function. (c) From the updated atomic species classification, we re-calculated the averaged intensity boxes for Pt and Pd and the resulting error functions to classify all the atoms again. This step was repeated until there was no change in the species classification, resulting in 11,474 Pt and 10,870 Pd atoms for Particle 1 (Revision Fig. 18b,c), and 3,435 Pt and 3,426 Pd atoms for Particle 2 (Revision Fig. 18e,f).

We have included this description in the ‘3D identification of atomic coordinates and chemical species’ part of the method section and included the figure of the integrated intensity histogram (Revision Fig. 18) as Supplementary Fig. 26.

Revision Figure 18 | Classification of chemical species. a-f, Histograms of the local intensities of all traced atoms (a,d), those of the atoms classified as Pd (b,e), and those of the atoms classified as Pt (c,f) for Particle 1 (a-c), and Particle 2 (d-f).

10. *The authors could briefly comment if there are some essential differences in the data acquisition and data processing compared to their paper Ref 43. The Ref. 43 also measures strain and calculates ORR.*

Response:

We thank the reviewer for the suggestion. Essential parts of our experiments and analyses are very similar to those of Ref. 43. There is a small difference in the ORR activity mapping; in this work, an additional facet assignment was performed in addition to the method described in Ref. 43 to map out the ORR activities from all surface atoms ('ORR activity calculation at the surface' part in the Methods). We have included this discussion in the 'ORR activity calculation at the surface' part of the method section, which now reads "Therefore, all surface atoms are assigned to one of the three dominant facet families (unlike our previous approach in which some surface atoms are excluded from ORR analysis).".

11. *SupplVideo2 and SupplVideo6 from "Zip of files for Reviewer" have low resolution and are highly compressed. It is not possible to distinguish blue and black dots.*

Response:

We thank the reviewer for the feedback. We have increased the resolution of all supplementary videos including Supplementary Videos 2 and 6.

12. *In SupplVideo4 and SupplVideo8 could have a legend of meaning of colors of the upper left particle. (It is described in the suppl. info, but label in the video would make it easier for reader to understand the video.)*

Response:

We thank the reviewer for the detailed suggestion. We added a legend to Supplementary Videos 4 and 8 to clarify the facet type.

References

1. Hata, S. *et al.* Electron tomography: An imaging method for materials deformation dynamics. *Current Opinion in Solid State and Materials Science* **24**, 100850 (2020).
2. Zhou, J. *et al.* Observing crystal nucleation in four dimensions using atomic electron tomography. *Nature* **570**, 500–503 (2019).
3. Vanrompay, H. *et al.* 3D characterization of heat-induced morphological changes of Au nanostars by fast *in situ* electron tomography. *Nanoscale* **10**, 22792–22801 (2018).
4. Skorikov, A. *et al.* Quantitative 3D Characterization of Elemental Diffusion Dynamics in Individual Ag@Au Nanoparticles with Different Shapes. *ACS Nano* **13**, 13421–13429 (2019).
5. Pryor, A. *et al.* GENFIRE: A generalized Fourier iterative reconstruction algorithm for high-resolution 3D imaging. *Sci Rep* **7**, 1–12 (2017).
6. Wise, A. M. *et al.* Inhibitive effect of Pt on Pd-hydride formation of Pd@Pt core-shell electrocatalysts: An *in situ* EXAFS and XRD study. *Electrochimica Acta* **262**, 27–38 (2018).
7. Davey, W. P. Precision Measurements of the Lattice Constants of Twelve Common Metals. *Phys. Rev.* **25**, 753–761 (1925).
8. Manchester, F. D., San-Martin, A. & Pitre, J. M. The H-Pd (hydrogen-palladium) System. *JPE* **15**, 62–83 (1994).
9. Dietze, E. M. & Grönbeck, H. Structure-Dependent Strain Effects. *ChemPhysChem* **21**, 2407–2410 (2020).
10. Yang, Y. *et al.* Atomic-scale identification of the active sites of nanocatalysts. <http://arxiv.org/abs/2202.09460> (2022) doi:10.48550/arXiv.2202.09460.
11. Lee, J., Jeong, C., Lee, T., Ryu, S. & Yang, Y. Direct Observation of Three-Dimensional Atomic Structure of Twinned Metallic Nanoparticles and Their Catalytic Properties. *Nano Lett.* **22**, 665–672 (2022).
12. An, H. *et al.* Atomic-scale structural identification and evolution of Co-W-C ternary SWCNT catalytic nanoparticles: High-resolution STEM imaging on SiO₂. *Science Advances* **5**, eaat9459.
13. Cui, C., Gan, L., Heggen, M., Rudi, S. & Strasser, P. Compositional segregation in shaped Pt alloy nanoparticles and their structural behaviour during electrocatalysis. *Nature Mater* **12**, 765–771 (2013).
14. Miyakawa, M. *et al.* Continuous syntheses of Pd@Pt and Cu@Ag core-shell nanoparticles using microwave-assisted core particle formation coupled with galvanic metal displacement. *Nanoscale* **6**, 8720–8725 (2014).

15. Li, J. & Sun, S. Intermetallic Nanoparticles: Synthetic Control and Their Enhanced Electrocatalysis. *Acc. Chem. Res.* **52**, 2015–2025 (2019).
16. Mimura, N., Hiyoshi, N., Daté, M., Fujitani, T. & Dumeignil, F. Microscope Analysis of Au–Pd/TiO₂ Glycerol Oxidation Catalysts Prepared by Deposition–Precipitation Method. *Catal Lett* **144**, 2167–2175 (2014).
17. Zhao, X. *et al.* Octahedral Pd@Pt_{1.8}Ni Core–Shell Nanocrystals with Ultrathin PtNi Alloy Shells as Active Catalysts for Oxygen Reduction Reaction. *J. Am. Chem. Soc.* **137**, 2804–2807 (2015).
18. Wang, X. *et al.* Palladium–platinum core-shell icosahedra with substantially enhanced activity and durability towards oxygen reduction. *Nat Commun* **6**, 7594 (2015).
19. Wu, Z. *et al.* Surface faceting and compositional evolution of Pd@Au core–shell nanocrystals during in situ annealing. *Physical Chemistry Chemical Physics* **21**, 3134–3139 (2019).
20. Kim, J. *et al.* Theoretical and Experimental Understanding of Hydrogen Evolution Reaction Kinetics in Alkaline Electrolytes with Pt-Based Core–Shell Nanocrystals. *J. Am. Chem. Soc.* **141**, 18256–18263 (2019).
21. Yang, Y. *et al.* Deciphering chemical order/disorder and material properties at the single-atom level. *Nature* **542**, 75–79 (2017).

Reviewer comments , second round review -

Reviewer #1 (Remarks to the Author):

Accept

Reviewer #2 (Remarks to the Author):

Thank you for the modification. Looks good to me.

Reviewer #3 (Remarks to the Author):

The revision addressed appropriately all my comments to the original version.

Please find below our response to the reviewers' comments on our manuscript "*Direct strain correlations at the single-atom level in three-dimensional core-shell interface structures*". We trust that the manuscript is now acceptable for publication in *Nature Communications*.

Response to reviewers' comments

Reviewer #1:

Accept

Response:

We thank the reviewer for taking the time and effort to provide constructive comments, which have been very helpful in making improvements to the manuscript.

Reviewer #2:

Thank you for the modification. Looks good to me.

Response:

We thank the reviewer for taking the time and effort to provide constructive comments, which have been very helpful in making improvements to the manuscript.

Reviewer #3:

The revision addressed appropriately all my comments to the original version.

Response:

We thank the reviewer for taking the time and effort to provide constructive comments, which have been very helpful in making improvements to the manuscript.